# Ascorbic Acid Induces the Increase of Secondary Metabolites, Antioxidant Activity, Growth, and Productivity of the Common Bean under Water Stress Conditions

**DOI:** 10.3390/plants9050627

**Published:** 2020-05-14

**Authors:** Alaa A. Gaafar, Sami I. Ali, Mohamed A. El-Shawadfy, Zeinab A. Salama, Agnieszka Sękara, Christian Ulrichs, Magdi T. Abdelhamid

**Affiliations:** 1Plant Biochemistry Department, National Research Centre, Cairo 12622, Egypt; aa.gaafar@nrc.sci.eg (A.A.G.); sb.ali@nrc.sci.eg (S.I.A.); zh.salama@nrc.sci.eg (Z.A.S.); 2Water Relations and Field Irrigation Department, National Research Centre, Cairo 12622, Egypt; elshawadfy77@yahoo.com; 3Department of Horticulture, University of Agriculture in Krakow, 31-425 Krakow, Poland; 4Division Urban Plant Ecophysiology, Faculty of Life Sciences, Humboldt-Universität zu Berlin, 14195 Berlin, Germany; christian.ulrichs@hu-berlin.de; 5Botany Department, National Research Centre, Cairo 12622, Egypt

**Keywords:** ascorbic acid, antioxidant systems, drought stress, *Phaseolus vulgaris*, secondary metabolites, yield

## Abstract

One of the most vital environmental factors that restricts plant production in arid and semi-arid environments is the lack of fresh water and drought stress. Common bean (*Phaseolus vulgaris* L.) productivity is severely limited by abiotic stress, especially climate-related constraints. Therefore, a field experiment in split-plot design was carried out to examine the potential function of ascorbic acid (AsA) in mitigating the adverse effects of water stress on common bean. The experiment included two irrigation regimes (100% or 50% of crop evapotranspiration) and three AsA doses (0, 200, or 400 mg L^−1^ AsA). The results revealed that water stress reduced common bean photosynthetic pigments (chlorophyll and carotenoids), carbonic anhydrase activity, antioxidant activities (2,2-diphenyl-1-picrylhydrazyl free radical activity scavenging activity and 2,2′-azino-bis(3-ethylbenzothiazoline-6-sulfonic acid) radical cation assay), growth and seed yield, while increased enzymatic antioxidants (peroxidase), secondary metabolites (phenolic, flavonoids, and tannins), malondialdehyde (MDA), and crop water productivity. In contrast, the AsA foliar spray enhanced all studied traits and the enhancement was gradual with the increasing AsA dose. The linear regression model predicted that when the AsA dose increase by 1.0 mg L^−1^, the seed yield is expected to increase by 0.06 g m^−2^. Enhanced water stress tolerance through adequate ascorbic acid application is a promising strategy to increase the tolerance and productivity of common bean under water stress. Moreover, the response of common bean to water deficit appears to be dependent on AsA dose.

## 1. Introduction

Water stress is one of the main stresses that after plant perception leads to an increase in the production of reactive oxygen species (ROS) [1]. ROS are able to injure the membrane of the cells and increase malondialdehyde (MDA) production [2]. However, both enzymatic and non-enzymatic systems are used by plants to scavenge over-produced ROS. The main enzymatic antioxidants involved in ROS detoxification processes are catalase (CAT), superoxide dismutase (SOD), ascorbate peroxidase (APX) and polyphenol oxidase (PPO) [3]. Plants produce a wide range of free radical scavenging molecules, including phenolic compounds, vitamins, and terpenoids [4,5,6]. Phenolic compounds are important components involved in plant growth and reproduction, and in the cell defense system against free radicals in conditions of abiotic and biotic stress [7].

Vitamins could be regarded as compounds of bioregulators or hormone precursors that, in tiny amounts, employ a valuable impact on plant growth and development. Overall, these substances could affect the energy metabolic pathway [8,9,10]. All essential physiological processes, such as photosynthesis, biosynthesis enzymes and secondary metabolites, nutrient and water absorption, and cell division more or less depend on vitamin availability. In the defense of plants in resistance to oxidative stress, vitamins with their antioxidant properties also play an important role as free radical scavengers. Vitamins have favourable effect on enhancing cell division and phytohormone synthesis, for instance cytokines and gibberellins [11]. Ascorbic acid (AsA) functions as an antioxidant, an enzyme cofactor, and as a pre-cursor for oxalate and tartrate synthesis. AsA is affiliated with chloroplasts in which the effect of oxidative stress on photosynthesis is mitigated. Furthermore, AsA abates the alteration of cell division and works as a primary substrate in the cyclic pathway of hydrogen peroxide enzymatic detoxification [12].

The common bean (*Phaseolus vulgaris* L.) is a grain legume with the world’s largest total production for direct human consumption, approximately 12 million tons per year, mostly in Latin America and Africa [13]. Beans are a source of protein, carbohydrates, fibre, minerals, and folate [14]. Environmental stresses, in particular climatic and soil related constraints severely restrict the yields of the common bean [15,16,17]. Fresh water is becoming ever more scarce due to climate change, particularly in arid environments, and causing droughts, which are becoming increasingly serious [18]. Fresh water scarcity is one of the most noteworthy environmental factors limiting crop production in arid environments [19,20,21,22]. The common bean is a rapidly growing plant that is very sensitive to soil-water conditions and soil quality, thus, the yield can be significantly decreased by even brief periods of water shortage [23]. Generally, water stress reduces the yield of the crop, but in a cultivar-dependent manner. Water management is an important factor in common bean production at all stages of plant development as it influences growth, yield, and yield quality. Water stress has been described to decrease many traits in faba beans except for the days to flowering and retention of moisture in the leaves [24,25].

Certain water management practices, e.g., irrigation can contribute to the sustained yield in low soil water environments; therefore, recognizing the effects of AsA on managing antioxidant metabolism systems and other physiological and biochemical processes is crucial in developing successful practices to enhance plant tolerance to water stress. The research hypothesis we verified is that ascorbic acid alleviates water stress tolerance of common bean plants and we can point physiological and biochemical determinants of this tolerance. The purpose of this work is also to examine the efficiency of foliar-applied AsA dose for counteracting the drought stress in common bean plants.

## 2. Results

### 2.1. Photosynthetic Pigments

The impacts of water management (WM) and ascorbic acid (AsA) on chlorophyll a (Chl a), chlorophyll b (Chl b), chlorophyll a + b (Chl a + b), and carotenoid (Car) content in common bean leaves are shown in Table 1. Water stress significantly reduced all mentioned parameters. Water stress decreased the total photosynthetic pigments (the sum of Chl a + b and Car) by 14.9% compared to the well-watered control. A similar trend was found regarding application of AsA, as it significantly affected all photosynthetic pigment traits, with a gradual increase with increasing AsA doses. Application of AsA at doses of 200 and 400 mg L^−1^ significantly increased the total photosynthetic pigments by 28.8% and 40.6%, respectively, compared to the well-watered control. The interaction effects of WM and AsA application revealed that water stress decreased photosynthetic pigments content in common bean leaves, while AsA foliar spray alleviated decrease in photosynthetic pigments content caused by the water deficit. AsA increased total photosynthetic pigments either in the control or water stress treatments. The increase in total photosynthetic pigments due to AsA treatment reached 39.0% in the control and 41.5% under the water stress treatment, at the higher AsA dose of 400 mg L^−1^ (Table 1).

### 2.2. Carbonic Anhydrase Activity

Water stress decreased the activity of carbonic anhydrase (CA) activity in common bean leaves by 43.7% compared to the well-watered control plants (Table 2). However, the application of AsA significantly increased carbonic anhydrase activity with a gradual increase with increasing AsA doses. Application of AsA significantly increased carbonic anhydrase activity by 23.2% and 25.1% in common bean plants sprayed with AsA at doses of 200 and 400 mg L^−1^, respectively, compared to the control. The interaction effects of WM and AsA application revealed that water stress reduced CA activity, but AsA foliar spray enhanced CA activity either in the well-watered control plants or under water stress treatments. The increase in CA activity due to AsA foliar spraying reached 13.4% in the well-watered control, and 50.9% under water stress treatment at the higher AsA dose of 400 mg L^−1^ (Table 2).

### 2.3. Antioxidant Enzyme Activities

Water stress increased the activity of peroxidase (POD) in common bean leaves by 86.8% compared to the well-watered control (Table 2). Similarly, the application of AsA significantly increased POD activity, with a gradual increase with increasing AsA doses. The application of AsA at doses of 200 and 400 mg L^−1^ increased POD activity by 15.0% and 31.9%, respectively compared to the control. The interaction effects of WM and AsA application revealed that not only water stress but also AsA foliar spray increased POD activity either in the leaves of common bean plants from the well-watered control or water stress treatments. The increase in POD activity due to AsA foliar spraying reached 27.2% in the well-watered control, and 34.8% under the water stress treatment at 400 mg L^−1^ AsA dose (Table 2).

### 2.4. Lipid Peroxidation

Water stress increased the malondialdehyde (MDA) accumulation in common bean leaves by 88.6% compared to the well-watered control (Table 2). The foliar application of AsA significantly decreased MDA with a gradual reduction with increasing AsA doses. Application of AsA significantly decreased MDA by 19.8% and 17.6% at doses of 200 and 400 mg L^−1^, respectively, compared to the control. The interaction effects of WM and AsA application revealed that though water stress increased lipid peroxidation reflected in MDA content, AsA foliar spray decreased it. AsA decreased lipid peroxidation either in the well-watered control plants or plants collected from the water stress treatments. The decrease in lipid peroxidation due to AsA foliar spraying reached 13.6% in well-watered control and 19.5% in plants under water stress treatment and a AsA dose of 400 mg L^−1^ AsA (Table 2).

### 2.5. Active Compounds

Water stress increased total phenolics (TP), total flavonoids (TF), and total tannins (TT) content in common bean leaves by 20.5%, 25.6%, and 11.1%, respectively, compared to the well-watered control (Table 3). The application of AsA at the dose of 400 mg L^−1^ significantly increased the TP, TF, and TT by 51.9%, 65.6%, and 54.7%, respectively, compared to the control. The increase in TP, TF, and TT due to AsA foliar application at the dose of 400 mg L^−1^ was higher than the increase resulting from the application of AsA at the dose of 200 mg L^−1^. The interaction effects of WM and AsA application revealed that not only water stress increased TP, TF, and TT content in common bean leaves but also AsA foliar spray had similar effect. The increase in TP due to AsA in the well-watered control plants reached 40.8% while it reached 62.4% under water stress treatment at the 400 mg L^−1^ AsA dose. The increase in TF due to AsA spraying reached 66.3% in the well-watered control while it reached 65.0% in the water stress treatment at the 400 mg L^−1^ AsA. The increase in TT due to AsA foliar spraying reached 51.8% in the well-watered control, and 58.3% in the water stress treatment at the 400 mg L^−1^ AsA dose (Table 3).

### 2.6. Identification of Phenolic Compounds by HPLC

The high-performance liquid chromatography (HPLC) phenolic profile of common bean plants is shown in Table 4. Methanol extract showed a remarkable content of pyrogallol (18.7 mg 100 g^−1^ DW), followed by kaempferol (6.96 mg 100 g^−1^ DW), hesperidin (5.76 mg 100 g^−1^ DW) and apigenin 6-arabinose 8-galactose (5.5 mg 100 g^−1^ DW) (Table 4). On the basis of our results, the free, bound, and total levels of these compounds were significantly affected by the ascorbic acid and irrigation regime, and their interactions. 

### 2.7. Antioxidant Activities (DPPH^•^ Radical Scavenging Activity and ABTS^•+^ Scavenging Activities)

Water stress decreased DPPH^•^ and ABTS^•+^ of common bean leaves by 10.3% and 10.0%, respectively, compared to the well-watered control (Table 5). The foliar application of AsA at a dose of 400 mg L^−1^ significantly increased DPPH^•^ and ABTS^•+^ activities by 37.7% and 34.3%, respectively, compared to the control treatment without AsA application. The increase in DPPH^•^ and ABTS^•+^ activity due to application of AsA at a dose of 400 mg L^−1^ was higher than the increase resulting from the application of the dose of 200 mg L^−1^. The interaction effects of WM and AsA application revealed that although water stress reduced DPPH^•^ and ABTS^•+^ activities, AsA foliar spray had adverse effect. AsA increased DPPH^•^ and ABTS^•+^ activity in the common bean leaves of both the well-watered control and water stress treatments. The increase in DPPH^•^ due to AsA reached 39.2% in the well-watered control and 35.4% under the water stress treatment at the 400 mg L^−1^ AsA dose. The increase in ABTS^•+^ activity due to AsA foliar spraying reached 33.4% in the plants of well-watered control and 35.7% under the water stress treatment at the AsA dose of 400 mg L^−1^ (Table 4).

### 2.8. Growth, Seed Yield and Crop Water Productivity

The impacts of water management and ascorbic acid on plant height (PH), leaf number per plant (LNo), leaf area per plant (LA), branch number per plant (BNo), seed yield (SY), and crop water productivity (WP) of common bean plants are shown in Table 6 and Table 7. Water stress significantly decreased PH, LNo, LA, BNo, and SY by 9.8%, 31.5%, 22.6%, 11.8%, and 27.2%, respectively, compared to the well-watered control. In contrast, water stress increased water productivity by 19.9% compared to the well-watered control. The foliar application of AsA significantly increased PH, LNo, LA, BNo, SY, and crop WP by 9.7%, 13.9%, 36.0%, 7.1%, 30.8%, and 32.4%, respectively, at AsA dose of 400 mg L^−1^, compared to the treatment without AsA application. The increase in common bean traits due to AsA dose of 400 mg L^−1^ was higher than the increase resulting from the application of 200 mg L^−1^ AsA. The interaction effects of WM and AsA application revealed that although water stress reduced PH, LNo, LA, BNo, and SY but the AsA foliar spray enhanced these traits. The increase in PH due to AsA in the well-watered control reached 8.1% and 11.5% under the water stress treatment at the AsA dose of 400 mg L^−1^. The increase in LNo per plant due to AsA foliar reached 15.4% for the well-watered control plants, and 12.6% under the water stress treatment at the AsA dose of 400 mg L^−1^. The increase in LA due to AsA application to the well-watered control reached 35.6% and 37.6% in the plants under the water stress treatment at the AsA dose of 400 mg L^−1^. The increase in BN due to AsA application in the well-watered control reached 5.1% and 9.7% under the water stress treatment at the AsA dose of 400 mg L^−1^. The increase in SY due to AsA spraying reached 25.9% for the plants of well-watered control and 38.7% for those of the water stress treatment at the AsA dose of 400 mg L^−1^. The increase in WP due to AsA foliar spraying reached 25.5% in the well-watered control and 38.7% under the water stress treatment at the AsA dose of 400 mg L^−1^ (Table 6 and Table 7).

### 2.9. Correlation Matrix

Pearson’s correlation coefficients (below diagonal) among all studied attributes of common bean plants grown under two water management treatments and three ascorbic acid doses are shown in Table 8. There was noted a strong correlation between seed yield and most of the studied traits, i.e., PH, LA, LNo, BNo, Chl a, Chll b, Chl a+b, Car, CA activity, DPPH^•^, and ABTS^•+^, which are highly positively associated with one another. There was no significant correlation between seed yield and TP or TF and TT. In addition, there was a negative association between seed yield and MDA, POD, and WP, notably that it was highly significant (*p* ≤ 0.01) only in the case of MDA.

### 2.10. Response Curve of Seed Yield to Ascorbic Acid Level

The linear response of seed yield (g m^−2^) to the ascorbic acid level (mg L^−1^) of common bean plants grown under two water management treatments and three ascorbic acid applications doses is shown in Figure 1. According to the model presented in Figure 1, if the AsA level increased by 1.0 mg L^−1^, the seed yield is expected to increase by 0.06 g m^−2^. The R^2^ value is the regression sum of squares divided by the total sum of squares. The estimated regression equation is significant at the 5% level of significance and about of 93.5% of the total variation in seed yield is explained by this equation.

### 2.11. Biplot Graph

Investigated treatments (green color) and measured traits (blue color) are shown in Figure 2, which shows a polygon view of ordination of treatment of main component analysis outputs by trait biplots. The biplot explained 97.25% of the total variation of the standardized data. The first and second principal components (PC1 and PC2) explained 65.88% and 31.37%, respectively. This comparatively high percentage reveals the complication of the relations among the treatments and the evaluated variables. T3 and T2 that coincided with AsA concentrations at a dose of 400 and 200 mg L^−1^ for the well-watered treatment scored significantly the highest values compared to other AsA concentrations for both water management treatments (well-watered and water-stressed) for PH, SY, LA, LNo, BNo, CA activity, ABTS^•+^, DPPH^•^, Chl a, Chl b, Chl a + b and Car. However, T5 and T6 scored the highest values in terms of POD and WP.

## 3. Discussion

Depending on the soil conditions, environment and variety, the total water requirement for a dry bean crop of 90−100 days may reach 350−500 mm h^−1^ [26]. Our results reported that 88 days were needed to complete the dry bean crop season with irrigation water application of 207.7 and 126 mm ha^−1^ season^−1^ for the well-watered control and water-stressed treatments, respectively, through sprinkler irrigation systems.

Mostly due to stomata closure and Rubisco inhibition, water stress resulted in a significant reduction of photosynthetic efficiency in plants [27]. Our results showed that reductions in chlorophyll a, chlorophyll b, chlorophyll a+b, and carotenoids under water stress conditions may be attributed to a mechanism of avoiding damage by reactive oxygen species (ROS) through the drop in leaf photosynthetic pigment content because of water deficits, as was stated by Herbinger et al. [28]. The carotenoid content decreased in a line with the chlorophyll content. The results of the present research confirmed Efeoğlu et al.’s [29] conclusion that the carotenoid content in plants was lower under water deficit conditions as a strategy of acclimation. Ascorbic acid (AsA) foliar application significantly increased chlorophyll a, chlorophyll b, carotenoids, and accordingly the total photosynthetic pigments in this study, especially at a dose of 400 mg L^−1^.

Previous research showed that exogenous AsA application under water stress increased chlorophyll a, chlorophyll b, and/or total chlorophyll contents [30,31]. In the mentioned studies, the increase in chlorophyll contents was mostly due to the protecting role of AsA under oxidative stress conditions. Moreover, photochemical efficiency depended on photosynthetic pigment contents, such as chlorophyll a and b, which influenced the photochemical reactions [32]. The increase of chlorophyll content affected by AsA application could also enhance the photochemical efficiency of common bean plants [33]. Our results are in conformity with those published by Hussein and Khursheed [34] showing that AsA protected photosynthesis and enhanced leaf photosynthetic pigments under drought conditions, controlling the dry matter accumulation.

The increase in carotenoid concentration in the common bean plants, with AsA application under water stress, protected plants against the damaging effect of ROS, which is completely bound to the core complexes of photosystem I and II, and was necessary for chloroplast performance. AsA is considered an efficient antioxidant, that participates in protecting photochemical processes [35]. Our results confirmed the findings of Helrich [36] and English [37], reporting that ascorbic acid application accelerates the scavenging of reactive oxygen species, promotes photosynthesis, and maintains enzyme activity at a stable level.

Assimilation of photosynthetic carbon was significantly reduced by 50% in water stress plants compared to those from the well-watered treatments in the present experiment. Water stress injured photosynthetic machinery at various levels e.g., pigments, stomatal performance, gas exchange, structure, and function of the thylakoid membrane, and electron transport regulated by the CO_2_ concentration [38,39]. AsA application increased carbonic anhydrase (CA) activity in common bean plants under water stress conditions. The present results are in harmony with those reported by Salama et al. [39] in *Phaseolus vulgaris* under conditions of salinity stress. AsA is a key compound in the activation of Rubisco, phosphoenolpyruvate carboxylase, and carbonic anhydrase under stress conditions [40]. This may be attributed to AsA-mediated alleviation in the activity of CA, which may mitigate the stress-dependent damage of the plasma membranes.

Plant defense mechanisms against water stress are also associated with peroxidase activity due to the peroxidase (POD) specific role in synthesis of phenols and saponins [41]. The results of the present study show that the POD activity in common bean leaves was significantly increased under conditions of water stress compared to well-watered plants. Thus, the POD activity significantly increased due to the oxidative stress induced by low water availability, as previously reported by Ramakrishna and Ravishankar [42]. As far as enzymatic activity was concerned, a significant increase was found in common bean leaves under water stress conditions. Externally-applied ascorbic acid enhanced POD activity, which is in accordance with the previous study on maize where POD and CAT activities have been increased due to exogenous application of ascorbic acid under water stress [43].

Water stress significantly enhanced the MDA content in common bean leaves. Exogenous application of AsA was effective in decreasing the MDA content under water stress conditions. Recent research by Elkeilsh et al. [44] confirmed that lipid peroxidation in wheat leaves increased due to water stress. Bor et al. [45] reported a significant increase in lipid peroxidation in sugar beet grown under water stress. Similar results were reported by Vaidyanathan et al. [46] in rice and Meloni et al. [47] in cotton. Malondialdehyde (MDA) is a marker of oxidative stress, which is a product of membrane lipid peroxidation, thus a higher MDA content should correspond to a higher degree of oxidative stress and is used to assess the degree of acclimatization of water-stressed plants [48]. Moreover, Cao et al. [49] reported that MDA concentration is good marker of structural integrity of membranes of plants cells exposed to water stress. A 50% higher MDA accumulation was determined in stressed common bean plants in conditions of a present research, indicating that plant cells were damaged by the water deficit, as it was also reported by Keyvan [48]. 

Water stress decreases plant growth by affecting various physiological and biochemical processes e.g., secondary metabolite synthesis, nutrient metabolism, respiration, and photosynthesis [50]. Production of phenols in plants commonly increases under environmental stress conditions [51,52]. Our results are in agreement with previous studies by Krol et al. [53], as they reported an increase in total phenolics, flavonoids, and tannins in common been, under conditions of water stress. In water-stressed plants, the production of flavonoids and other low-molecular-weight antioxidants may be partly correlated with leaf morphological variability and metabolic changes, which avoid oxidative injury [54]. Therefore, cells create certain responses, e.g., stress protein production, accumulation of organic solutes in different plant parts, and up regulation of antioxidant systems, e.g., antioxidant enzymes [55]. We found that application of ascorbic acid at a dose of 200 or 400 mg L^−1^ via foliar spray significantly increased total phenolics, total flavonoids, and total tannins under water stress conditions. These results confirmed those of Jaleel et al. [50] and Salama et al. [39] in *Phaseolus vulgaris* under salt stress.

In this study, two processes i.e., DPPH^•^ radical scavenging and an ABTS^•+^ assay were used to examine the antioxidant activity in the leaves of the common bean plants. Phenolic compounds are found to be associated with plant antioxidant activity [53]. Radical scavenging activity is also referred to phenolic compound structure i.e., the number and orientation of hydroxyl (−OH) groups [53]. Water stress leads to reactive oxygen species elevation, hence, a higher amount of antioxidants is needed to balance the stress environment and increase the plant adaptation processes [56]. Under water stress, the expression of several genes that contribute to the synthesis path of phenolic compounds e.g., phenylalanine ammonia-lyase is raised and needs high energy inputs, while these energy-intensive processes are restricted from moderate to severe stress environments [53]. In this study, significant increases were reported in antioxidant activity i.e. DPPH^•^ radical scavenging activity and ABTS^•+^ scavenging activity of common bean leaves due to ascorbic acid foliar spraying. The application of AsA caused the accumulation of metabolic regulators or bioactive compounds acting as cell component stabilizers which might alleviate the adverse effects of water stress. In addition, the further increase in DPPH^•^ activity and endogenous AsA was a result of strengthening the antioxidative defense system followed by an increased tolerance of bean plants to water stress. AsA may be engaged in the up-regulation of soluble sugars and proline synthesis to enhance tolerance mechanisms under water stress. The increase in the antioxidant activity was attributed to the increases of secondary metabolites (phenolics and flavonoids) which are considered as a tolerance advantage of water, as well as, other abiotic stresses as reported by and El-Amier et al. [52], Mostajeran and Rahimi Eichi [57], and Reddy et al. [58]. This study recommends the use of AsA at a dose of 200 or 400 mg L^−1^ for improving osmoprotectant and antioxidant systems of the common bean, with a positive growth and yield response as a consequence. 

In the present study, flavonoids and total phenolics content in common bean extracts are substantially based on the type of solvent, concentrations used, and also solvent polarity. These results were confirmed by Hounsome et al. [59] and Zhao et al. [60]. In general, the production of phenol compounds in plants under abiotic stress depended on type of stress, stress intensity, stress duration, plant development stages (usually, germination and plant development are the most sensitive to stress) and plant part type, i.e., whole seedlings or plant parts, namely roots or leaves [61,62]. The present results showed that water stress negatively affected common bean growth and yield traits, i.e., plant height, leaf number, leaf area, branch number, and seed yield. This may be attributed to a severe damage of a multitude of molecular, biochemical, and physiological processes, which adversely affected plant growth and development through reductions in cell elongation due to the inhibition of growth-promoting hormones, leading to a decrease in cell turgor and volume [63,64,65,66]. The response of a crops to water stress is determined by the species, genotype, water stress period and its severity, plant growth stage and its age, and the foliar-applied organic acid [65]. The exogenous application of AsA at 200 or 400 mg L^−1^ significantly enhanced all growth parameters in the light of the present results. Ultimate values of the growth parameters were obtained with the application of 400 mg L^−1^ AsA in the conditions of the present research. 

In stressed plants, ascorbic acid performs a crucial function in maintaining several metabolic processes [66]. Mittler [67] reported that AsA is one of the non-enzymatic antioxidant compounds serving as electron donors to reduce the accumulation of ROS and as a reaction substrate within the enzymatic cycle. These mechanisms are responsible for the differences between experimental treatments presented in this manuscript. It was reported that among a range of morphological, yield and related traits, seed yield determined in contrasting water management conditions (water stress and well watered) is considered the most efficient method for evaluating the integrated water stress tolerance in plants [68,69,70]. In addition, exogenous application of AsA improved plant performance under water stress and increased crop yield in maize (*Zea mays*) [71,72], flax (*Linum usitatissimum*) [73], and wheat (*Triticum aestivum*) [74]. 

Crop water productivity (WP) which is associated with the ratio of seed yield to water used, in general is inversely proportional to water stress severity. The WP varies with the plant growth stage that is influenced by the length of drought stress and its intensity. Huang et al. [75] reported that a high WP gave proof of the stomata’s function in sustaining high leaf water content during water stress. A high WP can lead to high photosynthetic activity. However, morphogenetic and metabolic studies are required to better understand the relation between stomatal closure and WUE. The reported WP values reached 3−6 kg ha^−1^ mm^−1^ [76]. In addition, Munoz-Perea et al. [77] reported that the WP values extended from 8.7−10.0 kg ha^−1^ mm^−1^. Our results showed that WP ranged from 4.63 to 7.86 kg ha^−1^ mm^−1^. Therefore, using agronomic practices such as an irrigation regime and exogenous application of ascorbic acid can enhance soil moisture conservation and WP would be a key for reducing water use and improving sustainable low-input production systems in the Nubaria region, Egypt and other water shortage environments of the world.

## 4. Materials and Methods 

### 4.1. Experimental Procedures

A field experiment was conducted during the 2017 season at the experimental station of National Research Centre, Nubaria, Egypt (30°86’67” N 31°16’67” E) withmean altitude 21 m above sea level). The experimental area is classified as an arid region with cool winters and hot dry summers. Daily temperatures ranged from 6.4−38.8 °C, with an average of 18.1 ± 4.1 °C. The daily relative humidity was 78.3 ± 9.1% in average, and ranged from 54−99%. Figure 3 shows the climatic data of the experimental site during the growing season. The mechanical analyses of the soil of the experimental site showed 68.9% coarse sand, 17.4% fine sand, 8.4% silt, and 5.3% clay, classified as sandy soil. 

The physical and chemical properties of the experimental soil are shown in Table 1, Table 2 and Table 3. Irrigation water was obtained from an irrigation channel passing through the experimental area. The chemical properties of the irrigation water are shown in Table 9. The experimental field was deeply plowed before planting. First disc harrow, then duck foot cultivator was used for further preparation of the field for planting. A combined driller that facilitated the concurrent application of fertilizer and seeds was used. 

Healthy common bean (*Phaseolus vulgaris* L.) cv. Nebraska seeds were sown on 29 September 2017 and harvested on 26 December 2017. Seeds were obtained from The Horticulture Research Institute, The Agricultural Research Centre, Giza, Egypt, and were sown at the amount of 95 kg ha^−1^ to achieve the recommended planting density.

Common bean seeds were selected for uniformity by choosing those of equal size and of the same color. The selected seeds were washed with distilled water, sterilized in 1% (*v/v*) sodium hypochlorite for approximately 2 min, washed thoroughly again with distilled water, and left to dry at room temperature (25 °C) for approximately 1 h. Uniform, air-dried common bean seeds were sown in hills in rows spaced 60 cm apart. The hills were spaced 10−15 cm apart in 3.0 m × 3.5 m plots. Thinning was done to produce two plants per hill. During soil preparation and plant growth, the soil was supplemented with the dose of NPK fertilizer according to the recommendations of the Ministry of Agriculture and Land Reclamation of Egypt for the studied area. These recommendations are 475 kg ha^−1^ of calcium super-phosphate (15.5% P_2_O_5_), 120 kg ha^−1^ ammonium sulfate (20.5% N), and 60 kg ha^−1^ potassium sulfate (48% K_2_O) during seed-bed preparation.

The experiment was conducted using a split-plot design in a randomized complete block design with three replicates. Specifically, irrigation as the first factor was assigned to the main plots. The main plots were then divided into sub-plots (split-plots). Next, dose of ascorbic acid (vitamin C) (Sigma-Aldrich, Saint Louis, MO, USA) as a second factor was allocated to the sub-plots and the number and timing of sprays were based on results from a preliminary pot trial (data not shown). The main plots included two irrigation regimes, namely, (1) 100% of crop evapotranspiration (ETc) throughout the season as a well-watered treatment (WW), and (2) 50% of ETc throughout the season as a water-stressed treatment (WS). The treatment of 50% ETc was applied to start from 19 days after sowing till the end of the growing season. The sub-plots were assigned to three ascorbic acid (AsA) treatments, i.e., (1) 0 mg L^−1^ AsA, (2) 200 mg L^−1^ AsA, and (3) 400 mg L^−1^ AsA. Ascorbic acid was applied to common bean plants as a foliar spraying, twice, at 25 and 40 day after sowing (DAS). All spray treatments were put on during the early morning and before 9:00 a.m., when wind is low, with a hand sprayer at sufficient pressure to keep droplet size small. Spray volume amount of water consisted of approximately 500 m^3^ per hectare. Plants were sprayed from both sides of the row in order to achieve adequate coverage. Therefore, the experiment consisted of six treatments as combinations of two irrigation regimes and three ascorbic acid doses, namely, T1—100% ETc with 0 mg L^−1^ AsA (WWAsA1); T2—100% ETc with 200 mg L^−1^ AsA (WWAsA2); T3—100% ETc with 400 mg L^−1^ AsA (WWAsA3); T4—50% ETc with 0 mg L^−1^ AsA (WSAsA1); T5—50% ETc with 200 mg L^−1^ AsA (WSAsA2); and T6—50% ETc with 400 mg L^−1^ AsA (WSAsA3). A preliminary experiment was carried out to verify which level of AsA could be used as foliar spraying. We concluded that the doses up to 400 mg L^−1^ are not harmful and can enhance plant growth. All other agricultural practices for common bean were carried out according to the recommendations of the Egyptian Ministry of Agriculture and Land Reclamation, Egypt.

### 4.2. Water Requirements and Irrigation Regimes 

Crop water requirements (CWR) was determined by estimating crop evapotranspiration (ETc) under standard conditions as follows:(1)ETc = ETo × Kc
where:

ETc = crop evapotranspiration [mm per day]

ETo = reference crop evapotranspiration [mm per day]

Kc = crop coefficient.

The values of ETc and CWR are equal, whereby ETc refers to the amount of water lost through evapotranspiration and CWR refers to the amount of water that is needed to compensate for the loss. ETc calculated from climatic data by direct integrating the effect of crop characteristics into ETo. The FAO Penman−Monteith method is recommended as the standard for calculating ETo. The Penman−Monteith equation is as follows [78]:(2)$${\mathrm{ET}}_{0}=\frac{0.408\Delta \;\left(R_{n}-G\right)+\gamma \frac{900}{T_{}+273}u_{2}\left(e_{s}^{}-e_{a}\right)}{\Delta +\gamma \left(1+0.34u_{2}\right)}$$
where: 

ET_o_ = reference evapotranspiration [mm per day] 

R_n_ = net radiation at the crop surface ([MJ m^−2^] per day) 

G = soil heat flux density ([MJ m^−2^] per day) 

T = mean daily air temperature at 2 m height [°C]

u_2_ = wind speed at 2 m height [m s^−1^]

e_s_ = saturation vapour pressure [kPa] 

e_a_ = actual vapour pressure [kPa]

e_s_−e_a_ = saturation vapour pressure deficit [kPa]

∆ = slope of saturation vapor pressure curve at temperature T [kPa °C] 

γ = psychrometric constant [kPa/ C].

The equation used the standard climatological records of solar radiation (sunshine), air temperature, humidity, and wind speed for daily calculations. The percentage of soil moisture content (*θ v*) was measured with the profile probe apparatus in sandy soils.

Amount of applied irrigation water (AW) was calculated according to the following equation for the sprinkler irrigation systems:(3)$$\mathrm{AW}=\frac{\mathrm{ETc}}{\mathrm{Ea}\times \left(1-\mathrm{LR}\right)}$$
where:

AW = applied irrigation water depth [mm per day]

Ea = application efficiency equals 75% for sprinkler irrigation system

LR = leaching requirements equals 10% for sprinkler irrigation system.

Irrigation time (IT) for the solid sprinkler system was calculated as follows:(4)$$\mathrm{Irrigation}\;\mathrm{time}\;\mathrm{in}\;\mathrm{hours}\;\left(\mathrm{IT}\right)=\frac{\mathrm{Applied}\;\mathrm{irrigation}\;\mathrm{water}}{\mathrm{Application}\;\mathrm{rate}\;\mathrm{for}\;\mathrm{sprinkler}\;\left(\mathrm{AR}\right)}$$
where:

AR = Application rate for sprinkler in [mm h^−1^]
(5)$$\mathrm{AR}=\frac{\mathrm{Sprinkler}\;\mathrm{number}\;x\;\mathrm{Sprinkle}\;\mathrm{discharge}\;\times 1000}{\mathrm{Strip}\;\mathrm{area}}$$
where:

Sprinkler discharge in [m^3^ h^–1^]

Strip area in [m^2^].

Crop water productivity (WP) is defined as the relationship between the seed yield and the amount of water elaborated in crop production. WP in kg mm^–1^ ha^–1^ was calculated as follows: (6)$$\mathrm{WP}=\mathrm{common}\;\mathrm{bean}\;\mathrm{seed}\;\mathrm{yield}\;\left({\mathrm{kg}\;\mathrm{ha}}^{-1}\right)/\mathrm{applied}\;\mathrm{water}\;\left({\mathrm{mm}\;\mathrm{ha}}^{-1}\right)$$

The irrigation regimes were started on the 18th of October in 2017 at 19 DAS and lasted for 70 days till the end of the season. The application of irrigation regimes through a sprinkler irrigation system were 207.7 and 126 mm ha^−1^ season^−1^ for 100% ETc (WW) and 50% ETc (WS), respectively. Plant samples were collected at 46 DAS for the determination of growth performance and biochemical analysis of common bean plants, where irrigation regimes lasted for 28 days. Notably, the common bean received 145.5 and 94.9 mm ha^−1^ sample^−1^ for 100% ETc (WW) and 50% ETc (WS), respectively.

### 4.3. Measurements 

Common bean plant samples were collected at 46 DAS during late vegetative growth and before flowering for the determination of growth performance and biochemical analysis. We carefully revmoved 108 plants, i.e., six plants from each plot × two levels of irrigation × three ascorbic acid levels × three replicate plots, from the experimental site, then dipped them in a bucket of water, and plants were shaken gently to remove all adhering soil particles.

Plant height (PH) in cm, leaf number per plant (LNo), leaf area per plant (LA) in cm^2^, and branch number per plant (BNo) of common bean plants were measured at 46 DAS, while seed yield (SY) in g m^−2^ and crop water productivity (WP) in kg mm^−1^ ha^−1^ of bean plants were measured at final harvest at 88 DAS.

Sample leaf disks were taken on the second fully-expanded not shaded leaf from the top using a cork borer 9 mm in diameter were used for determining chlorophyll, carotenoids, carbonic anhydrase activity (CA), peroxidase activity (POD), malondialdehyde (MDA), 2,2-diphenyl-1-picrylhydrazyl-free radical scavenging assay (DPPH^•^), and 2,2′-azino-bis(3-ethylbenzothiazoline-6-sulfonic acid) cation assay (ABTS^•+^). The leaf tissues that oven-dried for 72 h at 70 °C were used for estimating total phenolic (TP), total flavonoids (TF), and total tannins (TT). Common bean leaf chlorophyll a (Chl a), chlorophyll b (Chl b), and carotenoids (Car) concentrations were estimated using 80% (*v/v*) acetone extracts and the spectrophotometric method according to Lichtenthaler and Wellburn [79].

For preparation of enzyme extracts, about 5.0 g of plant material were crushed into a fine powder using liquid nitrogen. The sample was extracted by homogenizing the powder in 10 mL of 50 mM phosphate buffer (pH 7.8) containing 1 mM EDTA and 1% polyvinylpyrrolidine (PVP), with the addition of 1 mM ascorbate in the case of POD assay at 4 °C. The homogenate was centrifuged at 15,000× *g* for 20 min and the supernatant was used for the enzyme activity assay.

Carbonic anhydrase activity assay (CA; EC 4.2.1.1) were measured in leaves tissue (100 mg FW), which were homogenized with a buffered solution (pH 8.3) that contained 50 mm Veronal H_2_SO_4_ and 0.2% (*w/v*) PVP under ice cold-conditions. The homogenate was centrifuged at 12,000× *g* for 2 min and the supernatant was used for the determination of CA activity according to Ohki [80].

Peroxidase activity (POD; EC1.11.1.7) was assayed by monitoring the increase in absorbance at 430 nm due to the oxidation of pyrogallol [81]. The reaction mixture consisted of 50 mM potassium phosphate buffer (pH 7.0), 20 mM pyrogallol, 5 mM H_2_O_2_ and 20 µL of enzyme extract. In total, 1 unit of the enzyme was the amount necessary to decompose 1 µmoL of substrate per minute at 25 °C. Peroxidase activity was expressed as EU g^-1^ FW min^−1^. 

Lipid peroxidation was measured as the amount of malondialdehyde (MDA) determined by the thiobarbituric acid (TBA) reaction [82]. Frozen samples were homogenized with a pre-chilled mortar and pestle with two volumes of ice-cold 0.1% (*w/v*) trichloroacetic acid (TCA) and centrifuged for 15 min at 15,000× *g*. Assay mixture containing 1 mL of the supernatant and 2 mL of 0.5% (*w/v*) TBA in 20% (*w/v*) TCA was heated at 95 °C for 30 min and then rapidly cooled in an ice bath. After centrifugation (10,000× *g* for 10 min at 4 °C), the supernatant absorbance was read at 532 nm and the values corresponding to nonspecific absorption (600 nm) were subtracted. Lipid peroxidation products were measured as the content of TBA-reactive substances. MDA content was calculated according to the molar extinction coefficient of 155 mM cm^−1^.

ABTS^•+^ (2,2’-azinobis (3-ethylbenzothiazoline-6-sulfonic acid)), Folin-Ciocalteau reagents, gallic acid, quercetin, DPPH^•^ (2,2-diphenyl-1-picrylhydrazyl), BHT: butyl hydroxytoluene were purchased from Sigma Chemical Co. (St. Louis, MO, USA).

For preparation of common bean methanol extracts to determine secondary metabolites compounds and in vitro antioxidant activity, the dried powder of *P. vulgaris* leaves (10 g) was submersed separately in 100 mL methanol, for 24 h at room temperature using a shaker. Each mixture was filtered through Whatman No. 1 filter paper and this extraction step was repeated three times. The filtrate was then concentrated to dryness at 40 °C in a rotary evaporator. The crude extracts were stored in a refrigerator until chemical analysis.

The total phenolics (TP) were determined by Folin-Ciocalteu reagent assay using gallic acid as standard according to Singleton and Rossi [83]. A suitable aliquot (1 mL) of leaf extracts was added to 25 mL volumetric flask, containing 9 mL of distilled water. We added 1 mL of Folin-Ciocalteu’s phenol reagent to the mixture and shaken. After 5 min 10 mL of 7% Na_2_CO_3_ solution was added to the mixture. The solution was diluted to 25 mL with distilled water and mixed. After incubation for 90 min at room temperature, the absorbance was determined at 750 nm with a spectrophotometer (UV 300, Spectronic-Unicam, Cambridge, UK) against prepared reagent as blank. A total phenolic content in samples was expressed as mg of gallic acid equivalents (GAE) g^−1^ dry weight. All samples were analyzed in triplicates.

Total flavonoids (TF) were determined by the aluminum chloride method using quercetin as a standard [84]. We added 1 mL of plant extracts to 10 mL volumetric flasks, containing 4 mL of distilled water. To the flask 0.3 mL 5% NaNO_2_ was added and after 5 min 0.3 mL 10% AlCl_3_ was added. At 6 min, 2 mL 1 M NaOH were added and the total volume was made up to 10 mL with distilled water. The solutions were mixed well and the absorbance was measured against prepared reagent blank at 510 nm by using spectrophotometer UV 300. Total flavonoids in the sample were expressed as mg of quercetin equivalents (QE) g^−1^ fresh weight. Samples were analyzed in triplicates.

Total tannin (TT) was measured using the Folin-Ciocalteu reagent according to Polshettiwar et al. [85]. We added 1 mL of plant extracts to 7.5 mL distilled water (dH_2_O) then added 0.5 mL of Folin-Ciocalteu reagent and 1 mL of 35% sodium carbonate solution. The volume was made up for 10 mL with distilled water and absorbance was measured against prepared reagent blank at 775 nm by using spectrophotometer UV 300. Total tannins in the sample were expressed as mg of tannic acid equivalent (TE) g^−1^ dry weight. All samples were analyzed in triplicates.

Phenolics in mg 100 g^−1^ DW and flavonoids in mg 100 g^−1^ DW were measured by high-performance liquid chromatography (HPLC). A sample from 100% ETc with 400 mg L^−1^ (T3) treatment was used for measuring phenolics and flavonoids. Thus, the dried methanol crude extract (10 mg) of common bean was dissolved in 2 mL methanol HPLC spectral grade by vortex mixing for 15 min. The extract was filtrated through a 0.2 μm Millipore membrane filter. The phenolic and flavonoid compounds were identified by HPLC (Agilent Technologies 1100 series, Waldbronn, Germany), equipped with a quaternary pump (G131A model). The separation was achieved on the ODS reversed-phase column (C18, 25 × 0.46 cm i.d. 5 μm, Amsterdam, Netherlands). The injection volume (35 μL) was carried out with an auto sampling injector. The column temperature was maintained at 35 °C. Gradient phenolic compounds’ separation was carried out with an aqueous formic acid solution 0.1% (A) and methanol (B) as a mobile phase at a flow amount of 0.3 mL min^−1^ following the method of Goupy et al. [86]. In addition, the flavonoid compounds’ separation was carried out with 50 mM H_3_PO_4_, pH 2.5 (solution A) and acetonitrile (solution B) as a mobile phase at a flow rate of 0.7 mL min^−1^ as described by Mattila et al. [87]. Elutes were monitored using a UV detector set at 280 nm for phenolic acids, and at 330 nm for flavonoids. Chromatographic peaks were identified by comparing the retention times with the respective retention times of known standard reference material. Phenolic acid and flavonoid concentrations were calculated by comparing its peak areas with the peak areas of standards used (with known concentration) based on the data analysis of Hewlett Packard software. Phenolic acid and flavonoid compounds were expressed as µg 100 g^−1^ DW. 

Determination of DPPH^•^ (2,2-diphenyl-1-picrylhydrazyl) free radical scavenging activity at 75 µg mL^−1^ (DPPH%) was measured spectrophotometrically according to Chu et al. [88]. We prepared 0.1 mM of DPPH^•^ in methyl alcohol, and 0.5 mL of this solution was added to 1 mL of plant extract at a concentration (100 µg mL^−1^). Methanol was used as blank. The mixture was shaken vigorously and allowed to stand at room temperature. Butyl hydroxytoluene (BHT, Sigma) was used as a positive control; and negative control contained the entire reaction reagent except for the extracts. Then the absorbance was measured at 515 nm against the blank. Lower absorbance of the reaction mixture indicated higher free radical scavenging activity. The capacity to scavenge the DPPH^•^ radical was calculated using the following equation:(7)$${\mathrm{DPPH}}^{•}\;\mathrm{scavenging}\;\mathrm{activity}\;\%=\left[\left(A_{c}-A_{s}\right)∕Ac\right]\times 100$$
where: (A_c_) was the absorbance of the control reaction and (A_s_) the absorbance in the presence of the plant extracts.

ABTS^•+^ assay [2,2′-azino-bis(3-ethylbenzothiazoline-6-sulfonic acid)] enzymatic assay (ABTS%) was generated by oxidation of ABTS^•+^ with potassium persulphate [89]. ABTS^•+^ was dissolved in deionized water to 7.4 mM concentration, and potassium persulphate was added to a concentration of 2.6 Mm. The working solution was then prepared by mixing the two stock solutions in equal quantities and allowing them to react for 12−16 h at room temperature in the dark. The solution was then diluted by mixing 1 mL ABTS^•+^ solution with 60 mL methanol to obtain an absorbance of 1.1 ± 0.02 units at 734 nm using the spectrophotometer. Fresh ABTS^•+^ solution was prepared for each assay. Plant extract of 150 µL at a concentration of 100 µg mL^−1^ was allowed to react with 2850 µL of the ABTS^•+^ solution for 2 h in a dark condition. Then the absorbance was taken at 734 nm using the spectrophotometer. Results were expressed as in comparison with standard butylhydroxytoluene. A bigger antioxidant capacity of the sample exhibited a smaller production of free radicals. 

Percent activity was calculated using the equation:(8)$$\%\mathrm{Inhibition}=\left[\left(A_{0}-A_{1}\right)∕A_{0}\right]\times 100$$
where: A_0_ is the ABTS^•+^ absorbance of the control reaction and A_1_ is the ABTS^•+^ absorbance in the presence of the sample.

### 4.4. Statical Analysis

All data were subjected to analysis of variance for a split-plot design [90], after testing for the homogeneity of error variances using Levene test [91], and testing for normality distribution according to Shapiro and Wilk method [92]. Statistically significant differences between means were compared at *p* ≤ 0.05 using Tukey’s HSD (honestly significant difference) test. The statistical analysis was carried out using GenStat 17th Edition (VSN International Ltd, Hemel Hempstead, UK). Correlation coefficient *r* was calculated to determine the relationship between seed yield and each of the physiological and chemical traits. Hierarchical cluster analysis was performed on the standardized data using a measure of Euclidean distance and Ward minimum variance method as outlined by Ward [93]. Experimental data were also processed for a principal component analysis (PCA) using GenStat 17th Edition (VSN International Ltd, Hemel Hempstead, UK), in order to evaluate the existing relationships with original variables.

## 5. Conclusions

The harmful effects provoked by water stress could be alleviated by the exogenous application of ascorbic acid (AsA) in common bean plants. The key role of AsA was to enhance the common bean plants tolerance under water stress through increasing the antioxidative system capacity. AsA directly affected the reactive oxygen species (ROS) scavenging activity, promoted ROS scavenging enzyme activities (DPPH^•^ radical-scavenging activity and ABTS^•+^ scavenging activities), and increased the enzymatic antioxidants level (peroxidase). AsA application enhanced vegetative growth, seed yield, and water productivity in common bean plants. Regarding the biochemical and physiological level, AsA increased carbonic anhydrase activity, photosynthetic pigments (chlorophyll and carotenoids), and other secondary metabolites content (phenolic, flavonoids, and tannins), linked directly or indirectly with antioxidant properties. Additionally, the decrease in the malondialdehyde level in common bean leaves confirmed a protective role of AsA against cell membrane damage under water stress conditions. It was observed that AsA at a dose of 400 mg L^−1^, was, in most cases, the most effective in common bean plants subjected to water deficit regimes. Enhanced water stress tolerance through adequate ascorbic acid application is a promising strategy to improve water stress tolerance and the productivity of *P. vulgaris*.

## Figures and Tables

**Figure 1 plants-09-00627-f001:**
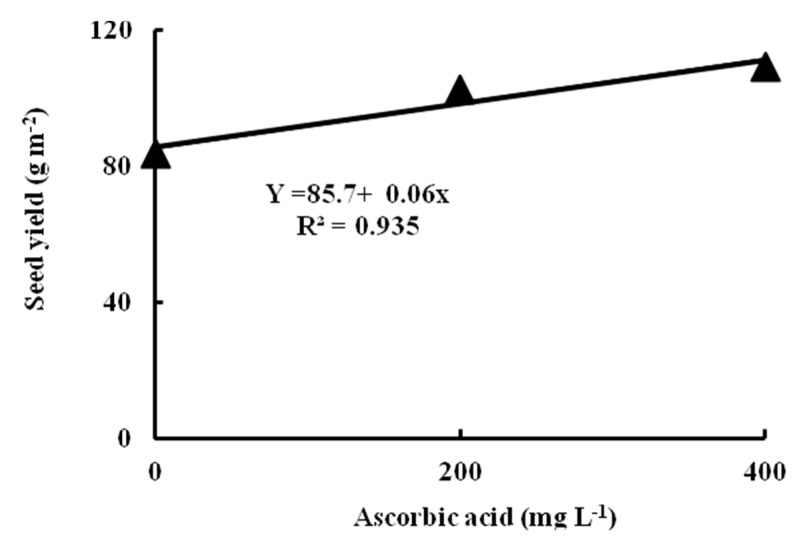
The response curve of common bean plants to ascorbic acid (AsA) applications amounts grown under two water management (WM) treatments.

**Figure 2 plants-09-00627-f002:**
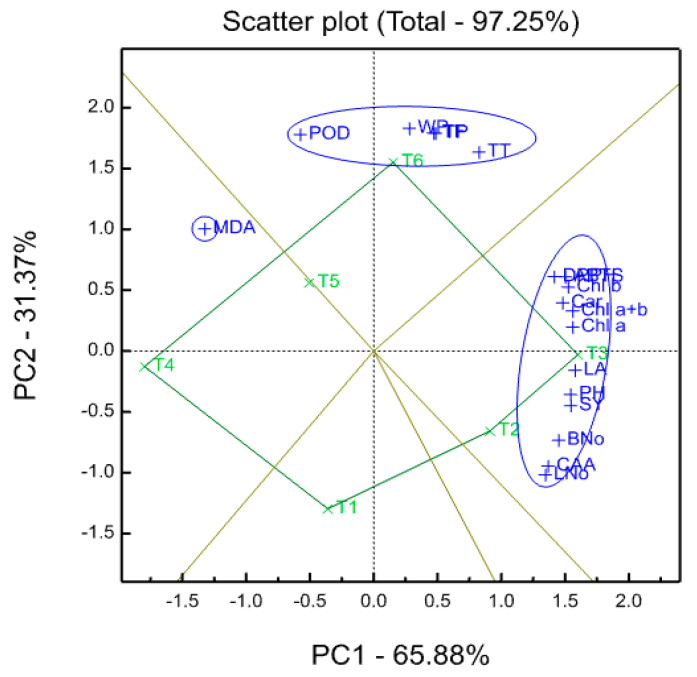
Ordination of treatment by trait biplots of principal component analysis outputs. Treatments in green colour and traits in blue colour. T1, 100% ETc with 0 mg L^−1^ AsA (WWAsA1); T2, 100% ETc with 200 mg L^−1^ AsA (WWAsA2); T3, 100% ETc with 400 mg L^−1^ AsA (WWAsA3); T4, 50% ETc with 0 mg L^−1^ AsA (WSAsA1); T5, 50% ETc with 200 mg L^−1^ AsA (WSAsA2); T6, 50% ETc with 400 mg L^−1^ AsA (WSAsA3). PH, plants plant height; LNo, leaf number per plant; LA, leaf area per plant; BNo, branch number per plant; Chl a, chlorophyll a; Chl b, chlorophyll b; Chl a+b, chlorophyll a+b; Car, carotenoids; CA, carbonic anhydrase activity; POD, peroxidase activity; MDA, malondialdehyde; TP, total phenolics; TF, total flavonoids; TT, total tannins; DPPH^•^, 2,2-diphenyl-1-picrylhydrazyl−free radical scavenging activity; ABTS^•+^, 2,2′-azino-bis(3−ethylbenzothiazoline-6-sulfonic acid) enzymatic assay; SY, seed yield; WP, crop water productivity.

**Figure 3 plants-09-00627-f003:**
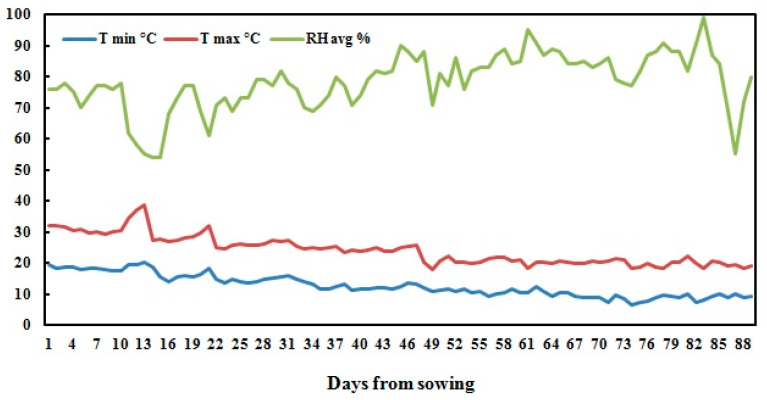
The data of maximum temperature, minimum temperature and relative humidity, obtained from weather station installed at the experimental station of National Research Centre, Nubaria region, Egypt.

**Table 1 plants-09-00627-t001:** The effect of water management (WM) and ascorbic acid (AsA) on photosynthesis pigments of chlorophyll a (Chl a), chlorophyll b (Chl b), chlorophyll a+b (Chl a+b), and carotenoids (Car) of common bean plants.

Treatment	Chl a	Chl b	Chl a + b	Car
	(mg g^−1^ FW)			
WM:				
WW	1.72 ^†^a	0.61a	2.33a	0.48a
WS	1.46b	0.49b	1.95b	0.44b
AsA:				
AsA1	1.32b	0.39b	1.71c	0.41c
AsA2	1.68a	0.58a	2.25b	0.47b
AsA3	1.78a	0.69a	2.46a	0.51a
WM × AsA:				
T1	1.42cd	0.47bc	1.89d	0.42c
T2	1.81ab	0.63ab	2.43ab	0.48b
T3	1.93a	0.73a	2.67a	0.55a
T4	1.21d	0.32c	1.53e	0.40c
T5	1.54c	0.52bc	2.07cd	0.45b
T6	1.62bc	0.64ab	2.26bc	0.47b

^†^ Mean values within the same column for each trait with the same lower case letter are not significantly different according to Tukey’s honestly significant difference (HSD) test at *p* ≤ 0.05. Measurements were done at 46 days after sowing. WW, well watered with 100% of evapotranspiration (ETc) throughout the season; WS, water stress, with 50% of ETc throughout the season; AsA1, 0 mg L^−1^ AsA; AsA2, 200 mg L^−1^ AsA; AsA3, 400 mg L^−1^ AsA; T1, 100% ETc with 0 mg L^−1^ AsA; T2, 100% ETc with 200 mg L^−1^ AsA; T3, 100% ETc with 400 mg L^−1^ AsA; T4, 50% ETc with 0 mg L^−1^ AsA; T5, 50% ETc with 200 mg L^−1^ AsA; T6, 50% ETc with 400 mg L^−1^ AsA.

**Table 2 plants-09-00627-t002:** The effect of water management (WM) and ascorbic acid (AsA) on carbonic anhydrase activity (CA) peroxidase activity (POD), and malondialdehyde (MDA) of common bean plants.

Treatment	CA	POD	MDA
	(EU g^−1^ FW)	(EU min^−1^ FW)	(µmol g^−1^ FW)
WM:			
WW	39.6 ^†^a	16.7b	19.3b
WS	22.3b	31.2a	36.4a
AsA:			
AsA1	26.7b	20.7c	31.8a
AsA2	32.9a	23.8b	25.5b
AsA3	33.4a	27.3a	26.2b
WM × AsA:			
T1	36.7b	14.7c	21.4c
T2	40.7ab	16.7c	18.0c
T3	41.6a	18.7c	18.5c
T4	16.7d	26.7b	42.1a
T5	25.1c	31.0b	33.1b
T6	25.2c	36.0a	33.9b

^†^ Mean values within the same column for each trait with the same lower-case letter are not significantly different according to Tukey’s HSD (honestly significant difference) test at *p* ≤ 0.05. Measurements were done at 46 day after sowing. WW, well watered with 100% of evapotranspiration (ETc) throughout the season; WS, water stress with 50% of ETc throughout the season; AsA1, 0 mg L^−1^ AsA; AsA2, 200 mg L^−1^ AsA; AsA3, 400 mg L^−1^ AsA; T1, 100% ETc with 0 mg L^−1^ AsA; T2, 100% ETc with 200 mg L^−1^ AsA; T3, 100% ETc with 400 mg L^−1^ AsA; T4, 50% ETc with 0 mg L^−1^ AsA; T5, 50% ETc with 200 mg L^−1^ AsA; T6, 50% ETc with 400 mg L^−1^ AsA.

**Table 3 plants-09-00627-t003:** The effect of water management (WM) and ascorbic acid (AsA) on total phenolics (TP), total flavonoids (TF) and total tannins (TT) of bean common plants.

Treatment	TP	TF	TT
	(mg g^−1^ DW)		
WM:			
WW	17.6 ^†^b	9.0b	3.61b
WS	21.2a	11.3a	4.01a
AsA:			
AsA1	15.4c	7.87c	2.96c
AsA2	19.5b	9.53b	3.89b
AsA3	23.4a	13.03a	4.58a
WM × AsA:			
T1	14.2e	6.85d	2.82e
T2	18.5c	8.74c	3.73c
T3	20.0b	11.39b	4.28b
T4	16.5d	8.89c	3.09d
T5	20.5b	10.32b	4.06b
T6	26.8a	14.67a	4.89a

^†^ Mean values within the same column for each trait with the same lower-case letter are not significantly different according to Tukey’s HSD (honestly significant difference) test at *p* ≤ 0.05. Measurements were done at 46 day after sowing. WW, well watered with 100% of evapotranspiration (ETc) throughout the season; WS, water stress with 50% of ETc throughout the season; AsA1, 0 mg L^−1^ AsA; AsA2, 200 mg L^−1^ AsA; AsA3, 400 mg L^−1^ AsA; T1, 100% ETc with 0 mg L^−1^ AsA; T2, 100% ETc with 200 mg L^−1^ AsA; T3, 100% ETc with 400 mg L^−1^ AsA; T4, 50% ETc with 0 mg L^−1^ AsA; T5, 50% ETc with 200 mg L^−1^ AsA; T6, 50% ETc with 400 mg L^−1^ AsA.

**Table 4 plants-09-00627-t004:** The HPLC phenolic profile of common bean plants from well watered treatment (100% ETc) with ascorbic acid dose of 400 mg L^−1^ sampled at 46 day after sowing.

Phenolics (mg 100 g^−1^ DW)	Methanol Extract	Flavonoids (mg 100 g^−1^ DW)	Methanol Extract
Pyrogallol	18.719	Apigenin 6-arbinose 8-galactose	5.500
Gallic acid	0.030	Narengin	0.640
3-OH tyrosol	0.649	Rosmarinic acid	0.155
Protocatchuic acid	-	Apigenin 6-rhamnose 8-glucose	0.089
Catechol	0.986	Luteolin 7-glucose	0.266
4-amino-benzoic	0.134	Hesperidin	5.763
Catechein	0.361	Rutin	0.108
Chlorogenic	0.417	Apigenin 7-glucose	0.051
*p*-H-enzoic acid	0.175	Apigenin 7-*O*-neohespiroside	0.165
Enzoic acid	1.600	Quercetrin	0.073
Caffeic acid	0.077	Naringenin	0.128
Vanillic acid	0.054	Quercetin	-
Caffeine	0.242	Hespirtin	-
*p*-coumaric acid	0.063	Acacetin 7-rutinoside	-
Ferulic	0.415	Kaempferol	6.964
Iso-ferulic acid	0.298	Apigenin	0.296
*α*-coumaric acid	0.042	-	-
Oleuropein	1.012	-	-
Coumarin	0.087	-	-
3,4,5-Methoxy-cinnamic acid	0.433	-	-

**Table 5 plants-09-00627-t005:** The effect of water management (WM) and ascorbic acid (AsA) on 2,2-diphenyl-1-picrylhydrazyl-free radical scavenging assay (DPPH%) at 75 µg mL^−1^ and 2,2′-azino-bis(3-ethylbenzothiazoline-6-sulfonic acid) enzymatic assay (ABTS%) of common bean plants.

Treatment	DPPH %	ABTS %
WM:		
WW	31.1 ^†^a	57.9a
WS	27.9b	52.1b
AsA:		
AsA1	25.2c	46.7c
AsA2	28.7b	55.5b
AsA3	34.7a	62.7a
WM × AsA:		
T1	26.8d	49.1e
T2	29.3c	58.9c
T3	37.3a	65.5a
T4	23.7e	44.2f
T5	28.1cd	52.2d
T6	32.1b	60.0b

^†^ Mean values within the same column for each trait with the same lower-case letter are not significantly different according to Tukey’s HSD (honestly significant difference) test at *p* ≤ 0.05. Measurements were done at 46 day after sowing. WW, well watered with 100% of evapotranspiration (ETc) throughout the season; WS, water stress with 50% of ETc throughout the season; AsA1, 0 mg L^−1^ AsA; AsA2, 200 mg L^−1^ AsA; AsA3, 400 mg L^−1^ AsA; T1, 100% ETc with 0 mg L^−1^ AsA; T2, 100% ETc with 200 mg L^−1^ AsA; T3, 100% ETc with 400 mg L^−1^ AsA; T4, 50% ETc with 0 mg L^−1^ AsA; T5, 50% ETc with 200 mg L^−1^ AsA; T6, 50% ETc with 400 mg L^−1^ AsA.

**Table 6 plants-09-00627-t006:** The effect of water management (WM) and ascorbic acid (AsA) on plant height (PH) (cm), leaf number per plant (LNo), leaf area per plant (LA), and branch number per plant (BNo) of common bean plants.

Treatment	PH	LNo	LA	BNo
	(cm)		(cm^2^ plant^−1^)	
WM:				
WW	28.5 ^†^a	17.8a	234a	4.07
WS	25.7b	12.2b	181b	3.59
				n.s.
AsA:				
AsA1	25.7b	13.7	172c	3.67
AsA2	27.4ab	15.7	216b	3.88
AsA3	28.2a	15.6	234a	3.93
		n.s.		n.s.
WM × AsA:				
T1	27.1ab	16.2ab	194b	3.93ab
T2	29.1a	18.5a	245a	4.13a
T3	29.3a	18.7a	263a	4.13a
T4	24.3b	11.1c	149c	3.40b
T5	25.8ab	12.9bc	187b	3.63ab
T6	27.1ab	12.5bc	205b	3.73ab

^†^ Mean values within the same column for each trait with the same lower-case letter are not significantly different according to Tukey’s HSD test at *p* ≤ 0.05, n.s.—not significant. Measurements were done at 46 day after sowing. WW, well watered with 100% of evapotranspiration (ETc) throughout the season; WS, water stress with 50% of ETc throughout the season; AsA1, 0 mg L^−1^ AsA; AsA2, 200 mg L^−1^ AsA; AsA3, 400 mg L^−1^ AsA; T1, 100% ETc with 0 mg L^−1^ AsA; T2, 100% ETc with 200 mg L^−1^ AsA; T3, 100% ETc with 400 mg L^−1^ AsA; T4, 50% ETc with 0 mg L^−1^ AsA; T5, 50% ETc with 200 mg L^−1^ AsA; T6, 50% ETc with 400 mg L^−1^ AsA.

**Table 7 plants-09-00627-t007:** The effect of water management (WM) and ascorbic acid (AsA) on seed yield (SY) and crop water productivity (WP) of common bean plants.

Treatment	SY	WP
	(g m−2)	(kg mm−1 ha−1)
WM:		
WW	114.2 ^†^a	5.49b
WS	83.0b	6.58a
AsA:		
AsA1	83.7b	5.09b
AsA2	102.5a	6.28a
AsA3	109.5a	6.74a
WM × AsA:		
T1	99.3b	4.78d
T2	118.3a	5.70cd
T3	125.0a	6.00bc
T4	68.0c	5.40cd
T5	86.7b	6.86ab
T6	94.3b	7.49a

^†^ Mean values within the same column for each trait with the same lower-case letter are not significantly different according to Tukey’s HSD (honestly significant difference) test at *p* ≤ 0.05. Measurements were done at 46 day after sowing. WW, well watered with 100% of evapotranspiration (ETc) throughout the season; WS, water stress with 50% of ETc throughout the season; AsA1, 0 mg L^−1^ AsA; AsA2, 200 mg L^−1^ AsA; AsA3, 400 mg L^−1^ AsA; T1, 100% ETc with 0 mg L^−1^ AsA; T2, 100% ETc with 200 mg L^−1^ AsA; T3, 100% ETc with 400 mg L^−1^ AsA; T4, 50% ETc with 0 mg L^−1^ AsA; T5, 50% ETc with 200 mg L^−1^ AsA; T6, 50% ETc with 400 mg L^−1^ AsA.

**Table 8 plants-09-00627-t008:** Pearson’s correlation coefficients (below diagonal) among all studied attributes of common bean plants grown under two water management and three ascorbic acid treatments.

Variables	PH	LNo	LA	BNo	Chl a	Chl b	Chl a + b	Car	CA	POD	MDA	TP	TF	TT	DPPH^•^	ABTS^•+^	WP
PH	1 ^†^																
LNo	0.921 ^**^	1															
LA	0.980 ^**^	0.888 ^*^	1														
BNo	0.975 ^**^	0.973 ^**^	0.932 ^**^	1													
Chl a	0.926 ^**^	0.783 ^ns^	0.976 ^**^	0.845 ^*^	1												
Chl b	0.883 ^*^	0.664 ^ns^	0.924 ^**^	0.778 ^ns^	0.963 ^**^	1											
Chl a + b	0.914 ^**^	0.742 ^ns^	0.964 ^**^	0.823 ^*^	0.995 ^**^	0.985 ^**^	1										
Car	0.829 ^*^	0.681 ^ns^	0.918 ^**^	0.727 ^ns^	0.951 ^**^	0.933 ^**^	0.956 ^**^	1									
CAA	0.933 ^**^	0.985 ^**^	0.885 ^*^	0.988 ^*^	0.782 ^ns^	0.699 ^ns^	0.754 ^ns^	0.667 ^ns^	1								
POD	−0.522 ^ns^	−0.797 ^ns^	−0.437 ^ns^	−0.677 ^ns^	−0.254 ^ns^	−0.087 ^ns^	−0.190 ^ns^	−0.160 ^ns^	−0.759 ^ns^	1							
MDA	−0.914 ^**^	−0.978 ^**^	−0.858 ^*^	−0.980 ^**^	−0.752 ^ns^	−0.667 ^ns^	−0.723 ^ns^	−0.622 ^ns^	−0.998**	0.773 ^ns^	1						
TP	0.151 ^ns^	−0.237 ^ns^	0.223 ^ns^	−0.051 ^ns^	0.384 ^ns^	0.556 ^ns^	0.452 ^ns^	0.424 ^ns^	−0.184 ^ns^	0.761 ^ns^	0.215 ^ns^	1					
TF	0.124 ^ns^	−0.251 ^ns^	0.214 ^ns^	−0.083 ^ns^	0.363 ^ns^	0.539 ^ns^	0.434 ^ns^	0.470 ^ns^	−0.209 ^ns^	0.746 ^ns^	0.252 ^ns^	0.972 ^**^	1				
TT	0.348 ^ns^	−0.012 ^ns^	0.446 ^ns^	0.151 ^ns^	0.602 ^ns^	0.734 ^ns^	0.659 ^ns^	0.651 ^ns^	0.029 ^ns^	0.608 ^ns^	0.010 ^ns^	0.959 ^**^	0.944 ^**^	1			
DPPH^•^	0.773 ^ns^	0.573 ^ns^	0.849 ^*^	0.661 ^ns^	0.883 ^*^	0.934 ^**^	0.914 ^**^	0.964 ^**^	0.595 ^ns^	−0.045 ^ns^	−0.547 ^ns^	0.528 ^ns^	0.594 ^ns^	0.719 ^ns^	1		
ABTS^•+^	0.859 ^*^	0.630 ^ns^	0.914 ^**^	0.735 ^ns^	0.956 ^**^	0.988 ^**^	0.977 ^**^	0.957 ^**^	0.647 ^ns^	−0.053 ^ns^	−0.606 ^ns^	0.583 ^ns^	0.590 ^ns^	0.758 ^ns^	0.955 ^**^	1	
WP	−0.004 ^ns^	−0.353 ^ns^	0.093 ^ns^	−0.190 ^ns^	0.288 ^ns^	0.442 ^ns^	0.351 ^ns^	0.325 ^ns^	−0.300 ^ns^	0.844 ^*^	0.322 ^ns^	0.955 ^**^	0.905 ^**^	0.918 ^**^	0.408 ^ns^	0.451 ^ns^	1
SY	0.992 ^**^	0.946 ^**^	0.983 ^**^	0.981 ^**^	0.927 ^**^	0.869 ^*^	0.911 ^**^	0.846 ^*^	0.953 ^**^	−0.567 ^ns^	−0.935 ^**^	0.085 ^ns^	0.070 ^ns^	0.306 ^ns^	0.780 ^*^	0.842 ^*^	−0.049 ^ns^

** and *, significant at 0.01 and 0.05 levels respectively; ns, non-significant. ^†^ Data used for calculating Pearson’s correlation coefficients are means of six treatments, which were replicated three times as a result of two water management treatments and three application rates of ascorbic acid. PH, plant height; LNo, leaf number per plant; LA, leaf area per plant; BNo, branch number per plant; Chl a, chlorophyll a; Chl b, chlorophyll b; Chl a + b, chlorophyll a + b; Car, carotenoids; CA, carbonic anhydrase activity; POD, peroxidase activity; MDA, malondialdehyde; TP, total phenolics; TF, total flavonoids; TT, total tannins; DPPH^•^, 2,2-diphenyl-1-picrylhydrazyl-free radical scavenging activity; ABTS^•+^, 2,2′-azino-bis(3-ethylbenzothiazoline-6-sulfonic acid) enzymatic assay; SY, seed yield; WP, crop water productivity.

**Table 9 plants-09-00627-t009:** Chemical and mechanical composition of extract of saturated soil of the experimental site and chemical characteristics of irrigation water used.

**Chemical Composition of Extract of Saturated Soil:**													
**Depth**	**pH**	**ECc**	**SP**		**CO_3_**	**HCO_3_**	**Cl^−^**	**SO_4_**		**Ca^2+^**	**Mg^2+^**	**Na^+^**	**K^+^**
(cm)	1:2.5	(dS m^−1^)											
0−20	8.13	1.90	22.1		-	2.00	12	4.30		5.20	4.10	8.27	0.73
20−40	8.1	1.88	21.0		-	1.00	13	2.80		5.08	4.00	7.20	0.52
40−60	8.09	1.96	21.0		-	1.30	12.2	4.65		5.00	3.00	9.61	0.54
**Elements Concentration of the Soil:**													
**N**	**P**	**K**	**Fe**		**Zn**	**Mn**	**Cu**						
(mg kg^−1^ soil)													
32.5	78.1	3.39	0.39		0.49	7.13	0.26						
**Mechanical Analyses of the Soil:**													
**Coarse sand**		**Fine sand**			**Silt**	**Clay**		**Texture**					
(%)													
68.9		17.4			8.4	5.3		Sandy					
**Chemical Characteristics of Irrigation Water:**													
**pH**	**EC**	**SAR**			**Anions and Cations (meq L^−1^)**								
					SO_-2_	Cl^-^	HCO^-^_3_	CO^-^_3_		K^+^	Na^+^	Mg^+2^	Ca^+2^
	(dS m^−1^)	(%)											
7.35	0.41	2.8			1.3	2.7	0.1	-		0.2	2.4	0.5	1.0

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
