# Peer review of "Ascorbic Acid Induces the Increase of Secondary Metabolites, Antioxidant Activity, Growth, and Productivity of the Common Bean under Water Stress Conditions"

_plants, 2020, doi:10.3390/plants9050627_

Round 1

Reviewer 1 Report

The manuscript „Ascorbic acid induces the increase of secondary metabolites, antioxidant activity, growth, and productivity of the common bean under water stress conditions” shows very promising results in terms of improving bean’s performance under drought stress. Nevertheless the paper itself needs a lot of improving to be published.

First thing to improve is the language. I am not a native English speaker but even I can spot some rather striking grammar errors.

Second thing that generally came to my mind – whe did the Authors use only two dosages of AsA? In most cases the higher dose gave better results, so it just asks for checking how far would that go. It should have a maximum level at which it helps, and then too concentrated AsA spray should rather harm the plants, at least I imagine it like that. Why was it not checked? Why wasn’t the optimal dosages established? I think that this study is incomplete without this.

Remarks to the particular fragments of the MS:

Abstract:

Line 35-37: I don’t think that such a detailed method description should be in this section.

Line 37-38: Here you should describe your results and not what was expected – whis could be stated earlier if it was a hypothesis (but was it?).

Introduction:

Line 59: what do you mean by „antioxidant possessions”?

Line 64: „alternation of cell division and protein” – protein what? Level? Structure? Expression? Location? There are many possibilities and the authors don’t use any. This is also a problem in the whole manuscript, frequently replicated in different situations.

Line 69-70: this sentence suggests that beans are a source of other legumes.

Line 79: what do you mean by „decrease the expression of many traits”?

Results:

Line 91: what do you mean by „reduced all mentioned traits”? The next sentence is also phrased awkwardly.

Line 97-98: what does it mean that it reduced the traits? And, more importantly, what does it mean that AsA spray improved them? Improved how? What does „improve” mean in this context? It is not a very scientific term.

Table 1: (and many other tables) – I do not think that putting a # at the and of the Table’s title is a good idea. I haven’t seen anything like that before. I suggest just putting it into the title (I mean the explanation for #), or putting the # in the Table, by the parameters measured.

Second thing about the Tables: you write in the Materials and Methods that there were six treatments, namely T1-T6 and that they are also referred to as WWAsA or WSAsA (1-3). And then we can see in the Tables results for WW, WS, AsA1-3 separately, and T1-T6 separately. What is WW then if not T1? What is AsA1, 2 and 3 if the foliar spray was always used on WW or WS plants? How can it be singled out? Where those results, that are different from T1-T6, come from?

Line 113-114: increased CA level? Activity? Accumulation? This is the same problem as earlier.

Line 117-118: again reduced traits and improved them – what does it mean?

Line 124-125 – the same as line 113-114, the same as line 128, 135, etc. (there is a lot more examples, but I will stop at that, the Authors should search for them themselves)

Line 180-181: Where are those results shown? Table 4 only gives us the level of those compounds 46 days after sowing. Was it WW or WS? Was it treated with AsA? We do not know.

Line 188: „a rate” implies a change in time, „level” would be more appropriate – this is used numerous times in the MS

Line 250: what do you mean by „evidence”? evidence for what? Did you mean correlation?

Line 253: „in a linear way” is not scientific language

Table 8: You wriet in the Table’s title that it refers to two WMs and three AsA treatments – but it is not shown in that Table. We can see various parameters but it is unclear when they were measured, in what conditions. The results certainly show one conditio though, not six, as described.

Line 263-269: As I wrote before, two dosages of AsA (plus control) is not enough for such conclusions – the Authors should test at least two more, and see if that tendency is real, and when it stops.

Paragraph 2.11 (Lines 273-286): I cannot see why it was done. These results have almost no relations to what was in the rest of the study and brings nothing to the conclusions. It looks like it was done just because it could be done and to increase the volume of the MS.

Paragraph 2.13 (Lines 320-338): First of all, this is a presentation of results that were alreade presented in a Table, which even the Authors admit. It is not in accordance with the rules of writing a scientific manuscript!

Secondly, again, here you cannot see different treatments (the same problem as in Table 4 or Table 8), only a comparison of the parameters but we do not know in what conditions. What is more, I do not see why this comparison was made (similar situation as for paragraph 2.11)

Discussion:

Line 342: application od what?

Line 344: it should be „results”, as this is a general statement

Line 346 and many other in the text: You should not be using „for” here

Line 376: what does it mean that AsA is central?

Line 378: what does it mean that it can correct the damage?

Line 384: what do you mean by „produced”?line 389-390: explain how oxidative stress is a product of membranÄ™ lipid peroxidation

Line 397-401: that shoud be earlier in this paragraph

Line 404: carbohydrates are not biochemical processes

Line 444-449: this is quite obvious, no need to write it in the discussion

Line 449-450: Well, changes in everything can be attributed to changes in gene expression, it i show the response to stress starts…

Line 450-452: is the response to water stress determined by exogenous application of organic acid though? It may be changed by it.

Line 469-470: this relation has been previously described though

Materials and Methods:

The authors should write which leaves (growth stages) were sampled for which analyses (did they sample and pool all of the leaves from a particular plant? Or did they choose e.g. first leaf for enzymes activity and second for something else?)

Line 493: Did you mean duckfoot cultivator?

Conclusions:

Line 716-717: what is the common bean plant status and what does it mean to improve it?

Author Response

Authors' Responses to Reviewer's Comments

Reviewer #1

The manuscript ʺAscorbic acid induces the increase of secondary metabolites, antioxidant activity, growth, and productivity of the common bean under water stress conditions” shows very promising results in terms of improving bean’s performance under drought stress. Nevertheless, the paper itself needs a lot of improving to be published.

Authors’ Response:

Thank you for a positive general opinion on the manuscript, and the detailed comments mentioned below. We carefully reviewed the manuscript and we hope that we suitably addressed your comments.

Reviewer #1

First thing to improve is the language. I am not a native English speaker but even I can spot some rather striking grammar errors.

Authors’ Response:

The text of the manuscript is after professional English proofreading by MDPI English Editing Service.

Reviewer #1

Second thing that generally came to my mind – whe did the Authors use only two dosages of AsA? In most cases the higher dose gave better results, so it just asks for checking how far would that go. It should have a maximum level at which it helps, and then too concentrated AsA spray should rather harm the plants, at least I imagine it like that. Why was it not checked? Why wasn’t the optimal dosages established? I think that this study is incomplete without this.

Authors’ Response:

We checked the available literature first before planning for this experiment. Besides, we carried out the preliminary experiment to verify the level of AsA we can apply as foliar spraying. From the preliminary experiment, we concluded that the doses up to 400 mg L-1 are not harmful and can enhance plant growth. Therefore, we did not need to use doses higher than 400 mg L-1. Also, the main purpose of this work was to examine the efficiency of foliar-applied AsA for counteracting the drought stress in common bean plants. Hence, pointing the safe level of AsA which brings the significant improvement of plant stress tolerance as compared to the control let to achieve the required target of this study.

Reviewer #1

Line 35-37: I don’t think that such a detailed method description should be in this section.

Authors’ Response:

We deleted these sentences.

Reviewer #1

Line 37-38: Here you should describe your results and not what was expected – whis could be stated earlier if it was a hypothesis (but was it?).

Authors’ Response:

Thank you for your note. This result is part of 2.10. The response curve of seed yield to the ascorbic acid level is the outcome of the linear regression analysis.

Reviewer #1

Line 59: what do you mean by ʺantioxidant possessions”?

Authors’ Response:

Sorry for this spelling error. We replaced ʺpossessionsʺ with the correct word “properties”.

Reviewer #1

Line 64: „alternation of cell division and protein” – protein what? Level? Structure? Expression? Location? There are many possibilities and the authors don’t use any. This is also a problem in the whole manuscript, frequently replicated in different situations.

Authors’ Response:

We deleted “and protein”.

Reviewer #1

Line 69-70: this sentence suggests that beans are a source of other legumes.

Authors’ Response:

We deleted “the other legumes”.

Reviewer #1

Line 79: what do you mean by ʺdecrease the expression of many traits”?

Authors’ Response:

We deleted “the expression of”.

Reviewer #1

Line 91: what do you mean by ʺreduced all mentioned traitsʺ? The next sentence is also phrased awkwardly.

Authors’ Response:

 “Water stress significantly reduced all mentioned traitsʺ.  This sentence is clear in the context of the previous sentence, where trials were mentioned. To clarify we replaced “traits” with “parameters”.

Reviewer #1

Line 97-98: what does it mean that it reduced the traits? And, more importantly, what does it mean that AsA spray improved them? Improved how? What does „improve” mean in this context? It is not a very scientific term.

Authors’ Response:

We clarified: “The interaction effects of WM and AsA application revealed that water stress decreased all photosynthetic pigments content, while AsA foliar spray alleviated decrease in photosynthetic pigments content caused by the water deficit”.

Reviewer #1

Table 1: (and many other tables) – I do not think that putting a # at the and of the Table’s title is a good idea. I haven’t seen anything like that before. I suggest just putting it into the title (I mean the explanation for #), or putting the # in the Table, by the parameters measured.

Authors’ Response:

Symbol # or any other symbol or even a number is used just after the title of the Tables to indicate important information below a table. Anyway, we delete this reference, according to the reviewer's suggestion.

Reviewer #1

Second thing about the Tables: you write in the Materials and Methods that there were six treatments, namely T1-T6 and that they are also referred to as WWAsA or WSAsA (1-3). And then we can see in the Tables results for WW, WS, AsA1-3 separately, and T1-T6 separately. What is WW then if not T1? What is AsA1, 2 and 3 if the foliar spray was always used on WW or WS plants? How can it be singled out? Where those results, that are different from T1-T6, come from?

Authors’ Response:

We have two factors the first water management (WM), which includes two levels i.e. well-watered treatment (WW), and water-stressed treatment (WS). The second factor is the application of ascorbic acid with three ascorbic acid (AsA) treatments, i.e., 0 mg L–1 (AsA1), 200 mg L–1 (AsA2), and 400 mg L–1 (AsA3). From these two factors (2 x 3) we got 6 treatments as a combination of the two factors as mentioned in the text and tables.

Reviewer #1

Line 113-114: increased CA level? Activity? Accumulation? This is the same problem as earlier.

Authors’ Response:

Carbonic anhydrase activity (CA) activity was measured in EU g–1 FW and this was mentioned in the text and the table.

Reviewer #1

Line 117-118: again reduced traits and improved them – what does it mean?

Authors’ Response:

Corrected

Reviewer #1

Line 124-125 – the same as line 113-114, the same as line 128, 135, etc. (there is a lot more examples, but I will stop at that, the Authors should search for them themselves)

Authors’ Response:

Corrected

Reviewer #1

Line 180-181: Where are those results shown? Table 4 only gives us the level of those compounds 46 days after sowing. Was it WW or WS? Was it treated with AsA? We do not know.

Authors’ Response:

Due to financial constraints to analyse several samples by HLPC, we analysed only the treatment which gave the most evident results on phenolic profile for common bean, i.e., T3, 100% ETc with 400 mg L–1 AsA.

Reviewer #1

Line 188: „a rate” implies a change in time, „level” would be more appropriate – this is used numerous times in the MS

Authors’ Response:

We replaced “rate” with “dose” according to reviewer’s comments and English editor suggestions.

Reviewer #1

Line 250: what do you mean by „evidence”? evidence for what? Did you mean correlation?

Authors’ Response:

We replaced “evidence” with “correlation”.

Reviewer #1

Line 253: „in a linear way” is not scientific language

We deleted “In a linear way”.

Reviewer #1

Table 8: You wriet in the Table’s title that it refers to two WMs and three AsA treatments – but it is not shown in that Table. We can see various parameters but it is unclear when they were measured, in what conditions. The results certainly show one conditio though, not six, as described.

Authors’ Response:

The correlation matrix is based on means. The number of means in the present experiment is 6: 2 factors with 3 levels = 6 treatments.

Reviewer #1

Line 263-269: As I wrote before, two dosages of AsA (plus control) is not enough for such conclusions – the Authors should test at least two more, and see if that tendency is real, and when it stops.

Authors’ Response:

Please, refer to any fundamental book in statistics, at least 3 values are necessary to use linear regression method and get its equation.

Reviewer #1

Paragraph 2.11 (Lines 273-286): I cannot see why it was done. These results have almost no relations to what was in the rest of the study and brings nothing to the conclusions. It looks like it was done just because it could be done and to increase the volume of the MS.

Authors’ Response:

The cluster analysis is important to explain or results. Is a technique to group similar observations into some clusters based on the observed values of several variables for each individual.

Reviewer #1

Paragraph 2.13 (Lines 320-338): First of all, this is a presentation of results that were alreade presented in a Table, which even the Authors admit. It is not in accordance with the rules of writing a scientific manuscript!

Authors’ Response:

We do not agree with the Reviewer’s opinion. Paragraph 2.13 (Lines 320-338), presents a vector view of the interrelationship among measured traits for six treatments resulting from the combination of water management and ascorbic acid, while Tables present the mean of each trait.

Reviewer #1

Secondly, again, here you cannot see different treatments (the same problem as in Table 4 or Table 8), only a comparison of the parameters but we do not know in what conditions. What is more, I do not see why this comparison was made (similar situation as for paragraph 2.11)

Authors’ Response:

W do not agree with the Reviewer’s opinion. It is well-known that the correlation matrix is based on the means. The number of means in the present experiment is 6: 2 factors and 3 levels = 6 treatments.

Reviewer #1

Line 342: application od what?

Authors’ Response:

Application of irrigation water. The L340-341 explains what you are asking about. In L340-341 it is written that reference (No.26) reports that they needed 350–500 mm h-1 , while we used (our results) 207.7 and 126 mm ha–1 season–1 for the well-watered control and water-stressed, respectively.

Reviewer #1

Line 344: it should be „results”, as this is a general statement

Authors’ Response:

We do not agree with the Reviewer’s opinion, that’s why we remain the text unchanged.

Reviewer #1

Line 346 and many other in the text: You should not be using „for” here

Authors’ Response:

Corrected.

Reviewer #1

Line 376: what does it mean that AsA is central?

Authors’ Response:

We replaced “central” with “key compound”

Reviewer #1

Line 378: what does it mean that it can correct the damage?

Authors’ Response:

We replaced “correct” with “mitigate”

Reviewer #1

Line 384: what do you mean by „produced”?line 389-390: explain how oxidative stress is a product of membranÄ™ lipid peroxidation

Authors’ Response:

We replaced “produced” with “induced”

Reviewer #1

Line 397-401: that shoud be earlier in this paragraph

Authors’ Response:

We changed the sequence of sentences in this paragraph.

Reviewer #1

Line 404: carbohydrates are not biochemical processes

Authors’ Response:

We deleted “carbohydrates”.

Reviewer #1

Line 444-449: this is quite obvious, no need to write it in the discussion

Authors’ Response:

It is the additional explanation of the results so we decided not to delete these sentences from the discussion chapter.

Reviewer #1

Line 449-450: Well, changes in everything can be attributed to changes in gene expression, it i show the response to stress starts…

Authors’ Response:

We deleted this sentence from the discussion chapter.

Reviewer #1

Line 450-452: is the response to water stress determined by exogenous application of organic acid though? It may be changed by it.

Authors’ Response:

We replaced the “exogenous application of an organic acid” with “the foliar-applied organic acid”

Reviewer #1

Line 469-470: this relation has been previously described though

Authors’ Response:

The context is different, that is the reason we decided to remain these sentences unchanged.

Reviewer #1

The authors should write which leaves (growth stages) were sampled for which analyses (did they sample and pool all of the leaves from a particular plant? Or did they choose e.g. first leaf for enzymes activity and second for something else?)

Authors’ Response:

Thanks for your note. We supplemented the necessary explanation in the text. Common bean plant samples were collected at 46 DAS during late vegetative growth and before flowering. Sample leaf disks were taken on the second fully-expanded unshaded leaf from the top using a  cork  borer  9  mm  in  diameter Fresh leaf tissues were used for determining chlorophyll, carotenoids, carbonic anhydrase activity (CA), peroxidase activity (POD), malondialdehyde (MDA), 2,2-diphenyl-1-picrylhydrazyl-free radical scavenging assay (DPPH•), and 2,2′-azino-bis(3-ethylbenzothiazoline-6-sulfonic acid) cation assay (ABTS•+).

Reviewer #1

Line 493: Did you mean duckfoot cultivator?

Authors’ Response:

Yes, we mean Duck foot cultivator. We corrected “food” to “foot”. It was just a spelling error.

Reviewer #1

Conclusions:

Line 716-717: what is the common bean plant status and what does it mean to improve it?

Authors’ Response:

We replaced the “status” with “tolerance”, and “improve” with “increase”.

Reviewer 2 Report

General remarks. This study analyses influence of ascorbic acid on accumulation of biologically active substances, antioxidant activity and growth of Phaseolus vulgaris L. The novelty of the research was presented in this manuscript.  On the other hand, there are some inaccuracies in experiments and data analysis.

The title of the manuscript reflects the content and abstract provides information on the studies performed. The keywords are appropriate. Abstract is adequate and contains short information on methods, results and conclusions.

The introduction chapter justify novelty of the research presented in the manuscript. Authors analyze references and formulate the purpose in this chapter. Since there are a number of research variants in this study, authors should also make hypotheses which could to explain predictions of the relationships among variables.

Materials and Methods.  Authors did not specify the control that is required for field experiments. Statistical significant differences were compared using Tukey’s test so interpretation of the results in Tables 1 (line101), Table 2 (line 143), Table 3 (line 167), Table 5 (line 196), Table 6 (line 230), Table 7 (line 239) was confusing. This chapter should clearly present the conception of control with which test variants were compared.

Results. The results are described too extensive without focusing on the essentials with as many as eight tables and four graphs. In my opinion, the same results are interpreted several times using different methods statistical analysis. It is unclear what indicates a negative correlation in Section 2.11: Figs. 2 or Table 8 discussed here. The authors should reject any of the figures in Figs. 2-4. Thus, it is recommended to present the results in a very specific manner, highlighting the most significant differences.

Discussion. The results obtained are discussed properly by comparison with the studies of other authors.

Conclusions. Since no hypotheses have been formulated, the results are simply listed, so the conclusions are descriptive in substance.

Author Response

Reviewer #2

General remarks. This study analyses influence of ascorbic acid on accumulation of biologically active substances, antioxidant activity and growth of Phaseolus vulgaris L. The novelty of the research was presented in this manuscript.  On the other hand, there are some inaccuracies in experiments and data analysis.

Authors’ Response:

Thank you for a positive general opinion on the manuscript, and detailed comments mentioned below. We carefully reviewed the manuscript and we hope that we satisfactorily referred your comments.

Reviewer #2

The title of the manuscript reflects the content and abstract provides information on the studies performed. The keywords are appropriate. Abstract is adequate and contains short information on methods, results and conclusions.

Authors’ Response:

Thank you for a positive general opinion on the manuscript, and detailed comments mentioned below.  

Reviewer #2

The introduction chapter justify novelty of the research presented in the manuscript. Authors analyze references and formulate the purpose in this chapter. Since there are a number of research variants in this study, authors should also make hypotheses which could to explain predictions of the relationships among variables.

Authors’ Response:

Thanks for your note. We added the detailed hypothesis of the study. Besides, we presented a linear response curve that allows predicting seed yield under given conditions.

Reviewer #2

Materials and Methods.  Authors did not specify the control that is required for field experiments. Statistical significant differences were compared using Tukey’s test so interpretation of the results in Tables 1 (line101), Table 2 (line 143), Table 3 (line 167), Table 5 (line 196), Table 6 (line 230), Table 7 (line 239) was confusing. This chapter should clearly present the conception of control with which test variants were compared.

Authors’ Response:

We used negative control as a base for comparison of treatments. It is more common now than the use of positive control, but we understand and consider your note. Tukey's test controls the experiment error rate, so, we used Tukey's HSD (honestly significant difference) test for testing the statistically significant differences between means compared at P ≤ 0.05.

Reviewer #2

Results. The results are described too extensive without focusing on the essentials with as many as eight tables and four graphs. In my opinion, the same results are interpreted several times using different methods statistical analysis. It is unclear what indicates a negative correlation in

Authors’ Response:

We presented the means and marginal means in Tables with Pearson’s correlation coefficients as well, while figures clarified the results through a different approach. Figure 1 illustrates the response curve of common bean plants to ascorbic acid (AsA) application rates, while Figure 2 showed the extent of distances among different evaluated variables. Besides, Figure 3 & 4 showed the principal component analysis biplot, which displays the examined 18 variables tested under 6 treatments as a combination of the two factors water management (WM), and three ascorbic acid (AsA) treatments.

Reviewer #2

Section 2.11: Figs. 2 or Table 8 discussed here. The authors should reject any of the figures in

Authors’ Response:

We deleted Fig. 2.

Reviewer #2

Figs. 2-4. Thus, it is recommended to present the results in a very specific manner, highlighting the most significant differences.

Authors’ Response:

We carefully reviewed the manuscript and we hope that we adequately referred your comments.

Reviewer #2

Discussion. The results obtained are discussed properly by comparison with the studies of other authors.

Authors’ Response:

Thank you for a positive opinion on the Discussion.

Reviewer #2

Conclusions. Since no hypotheses have been formulated, the results are simply listed, so the conclusions are descriptive in substance.

Authors’ Response:

Thanks for your note. We added a hypothesis and provided in the introduction section.

Reviewer 3 Report

The manuscript "Ascorbic acid induces the increase of secondary metabolites, antioxidant activity, growth, and productivity of the common bean under water stress conditions" describes how using the ascorbic acid can improve  biochemical/physiological traits of water stress-grown common bean. 

The biggest concern about the research refers to the methodology. 

The Authors used 2 water regimes and 3 ascorbic acid doses and then their interaction. That is, in general correct;

however, the Authors  made a serious mistake in the design. They did not explain what water regime was used under "0" AsA (named in the text as "AsA 1"). Moreover, the "T1" treatment (0 AsA and 100% water) is exactly the same as control WW (100% water); and similar: the "T4" treatment had 0 AsA and 50% of water--so it was exactly the same as WM treatment (50% of water only).

Having the mistake in the original research design, the further statistics and thus results are erroneous.

The Authors are strongly recommended to consider the whole plot design again and correct it. Maybe excluding some "ghost" treatments would help.

Author Response

Reviewer #3

The manuscript "Ascorbic acid induces the increase of secondary metabolites, antioxidant activity, growth, and productivity of the common bean under water stress conditions" describes how using the ascorbic acid can improve  biochemical/physiological traits of water stress-grown common bean.

Authors’ Response:

Thank you for a positive general opinion on the manuscript, and detailed comments mentioned below.  

Reviewer #3

The biggest concern about the research refers to the methodology. The Authors used 2 water regimes and 3 ascorbic acid doses and then their interaction. That is, in general correct; however, the Authors  made a serious mistake in the design.

They did not explain what water regime was used under "0" AsA (named in the text as "AsA 1").  Moreover, the "T1" treatment (0 AsA and 100% water) is exactly the same as control WW (100% water); and similar: the "T4" treatment had 0 AsA and 50% of water--so it was exactly the same as WM treatment (50% of water only).

Having the mistake in the original research design, the further statistics and thus results are erroneous. The Authors are strongly recommended to consider the whole plot design again and correct it. Maybe excluding some "ghost" treatments would help.

Authors’ Response:

Thanks for your note. Please, refer to M & M section. Please, note that we had two irrigation regimes, namely, 1) 100% of crop evapotranspiration (ETc) throughout the season as a well-watered treatment (WW), and 2) 50% of ETc throughout the season as a water-stressed treatment (WS). In addition we had three ascorbic acid (AsA) treatments, i.e., 1) 0 mg L–1 AsA, 2) 200 mg L–1 AsA, and 3) 400 mg L–1 AsA. Therefore, the experiment consisted of six treatments as combinations of two irrigation regimes and three ascorbic acid doses,  namely,  T1, 100% ETc with 0 mg L–1 AsA (WWAsA1); T2, 100% ETc with 200 mg L–1 AsA (WWAsA2); T3, 100% ETc with 400 mg L–1 AsA (WWAsA3); T4, 50% ETc with 0 mg L–1 AsA (WSAsA1); T5, 50% ETc with 200 mg L–1 AsA (WSAsA2); and T6, 50% ETc with 400 mg L–1 AsA (WSAsA3).

Reviewer #3

They did not explain what water regime was used under "0" AsA (named in the text as "AsA 1"). 

Authors’ Response:

Under “0” AsA it was two treatments i.e., T1, 100% ETc with 0 mg L–1 AsA (WWAsA1); and T4, 50% ETc with 0 mg L–1 AsA (WSAsA1).

Reviewer #3

Moreover, the "T1" treatment (0 AsA and 100% water) is exactley the same as control WW (100% water); and similar: the "T4" treatment had 0 AsA and 50% of water--so it was exactly the same as WM treatment (50% of water only).

Authors’ Response:

Please, note that “T1, treatment (0 AsA and 100% water)”not same as “control WW (100% water)”. The “control WW (100% water)” is level one of water management treatments, and not  a treatment. In addition, the "T4, had 0 AsA and 50% of water” not same as “WM treatment (50% of water only)”. The 50% of water only is the second level of water management factor and not treatment. The Treatment is “0 AsA and 50% of water” which is T4.

Round 2

Reviewer 1 Report

Reviewer #1

Second thing that generally came to my mind – whe did the Authors use only two dosages of AsA? In most cases the higher dose gave better results, so it just asks for checking how far would that go. It should have a maximum level at which it helps, and then too concentrated AsA spray should rather harm the plants, at least I imagine it like that. Why was it not checked? Why wasn’t the optimal dosages established? I think that this study is incomplete without this.

Authors’ Response:

We checked the available literature first before planning for this experiment. Besides, we carried out the preliminary experiment to verify the level of AsA we can apply as foliar spraying. From the preliminary experiment, we concluded that the doses up to 400 mg L-1 are not harmful and can enhance plant growth. Therefore, we did not need to use doses higher than 400 mg L-1. Also, the main purpose of this work was to examine the efficiency of foliar-applied AsA for counteracting the drought stress in common bean plants. Hence, pointing the safe level of AsA which brings the significant improvement of plant stress tolerance as compared to the control let to achieve the required target of this study.

Re: Then the information about it should be included in the introduction, or in the Materials and metods section, not only in this response.

Reviewer #1

Second thing about the Tables: you write in the Materials and Methods that there were six treatments, namely T1-T6 and that they are also referred to as WWAsA or WSAsA (1-3). And then we can see in the Tables results for WW, WS, AsA1-3 separately, and T1-T6 separately. What is WW then if not T1? What is AsA1, 2 and 3 if the foliar spray was always used on WW or WS plants? How can it be singled out? Where those results, that are different from T1-T6, come from?

Authors’ Response:

We have two factors the first water management (WM), which includes two levels i.e. well-watered treatment (WW), and water-stressed treatment (WS). The second factor is the application of ascorbic acid with three ascorbic acid (AsA) treatments, i.e., 0 mg L–1 (AsA1), 200 mg L–1 (AsA2), and 400 mg L–1 (AsA3). From these two factors (2 x 3) we got 6 treatments as a combination of the two factors as mentioned in the text and tables.

Re: Okay, and I understood it that way the forst time I read your MS, but this is not what I was asking about. In Materials and Methods you wrote: „the experiment consisted of six treatments as combinations of two irrigation regimes and three ascorbic acid doses, namely, T1 – 100% ETc 515 with 0 mg L–1 AsA (WWAsA1); T2 – 100% ETc with 200 mg L–1 AsA (WWAsA2); T3 – 100% ETc with 516 400 mg L–1 AsA (WWAsA3); T4 – 50% ETc with 0 mg L–1 AsA (WSAsA1); T5 – 50% ETc with 200 mg 517 L–1 AsA (WSAsA2); and T6 – 50% ETc with 400 mg L–1 AsA (WSAsA3).”

I asked how did you single out only WW, only WS, only AsA1-3? It is inconsistent with Materials and Methods and I don’t know still where it came from. You write that you had 6 treatment combinations and I asked what is AsA1 (alone) if it is now WWAsA1 (T1)? What is WS (alone) if it is not WSAsA1 (T4)? I don’t understand how you got the results for AsA1 that are different than the results for T1. Is AsA1 a mean for T1 and T4 or something? If it is, and you got the results for single factors by any kind of calculations, than it should be stated clearly in the text, also in the tables’ descriptions.

Line 180-181: Where are those results shown? Table 4 only gives us the level of those compounds 46 days after sowing. Was it WW or WS? Was it treated with AsA? We do not know.

Authors’ Response:

Due to financial constraints to analyse several samples by HLPC, we analysed only the treatment which gave the most evident results on phenolic profile for common bean, i.e., T3, 100% ETc with 400 mg L–1 AsA.

Re: it should be indicated in the Materials and Methods, that it was only measured for this time point then.

Reviewer #1

Table 8: You write in the Table’s title that it refers to two WMs and three AsA treatments – but it is not shown in that Table. We can see various parameters but it is unclear when they were measured, in what conditions. The results certainly show one conditio though, not six, as described.

Authors’ Response:

The correlation matrix is based on means. The number of means in the present experiment is 6: 2 factors with 3 levels = 6 treatments.

Re: So you show a mean for all 6 treatments for one parameter? For example chlb in this table is a mean value received for chlb content in all T1-T6 treatments? If so, then it should be clearly stated in the description of the table, and in Materials and Methods, so that you will not be misunderstood (like you were now by me). Secondly I do not see why the authors put those results in this study. They are interesting but the correlations between various parameters without context (the context was lost by calculating the mean for all of the treatments, now this is only a correlation between those parameters in bean) should be put in a different study, with different research hypothesis.

Reviewer #1

Paragraph 2.11 (Lines 273-286): I cannot see why it was done. These results have almost no relations to what was in the rest of the study and brings nothing to the conclusions. It looks like it was done just because it could be done and to increase the volume of the MS.

Authors’ Response:

The cluster analysis is important to explain or results. Is a technique to group similar observations into some clusters based on the observed values of several variables for each individual.

Re: I know what cluster analysis is, and what it is used for. I just don’t see the connection between the aim of this study, your research hypothesis, and showing cluster analysis for the parameters if it is done on means from all of the treatments. You lose the connection to your study in that way and get the results which are interesting, but I do not see how they are important to explain your results. They would be if they were shown separately for each treatment x treamtment correlation.

Reviewer #1

Paragraph 2.13 (Lines 320-338): First of all, this is a presentation of results that were alreade presented in a Table, which even the Authors admit. It is not in accordance with the rules of writing a scientific manuscript!

Authors’ Response:

We do not agree with the Reviewer’s opinion. Paragraph 2.13 (Lines 320-338), presents a vector view of the interrelationship among measured traits for six treatments resulting from the combination of water management and ascorbic acid, while Tables present the mean of each trait.

Re: Yes, the Table presents the correlations (interrelationship) between means of the measured traits, and the Figure presents a vector view of the interrelationship (correlation) among means of the measured traits – I still think you should only pick one. You can observe that the parameters with correlation close to 1 are very close to each other in this figure etc. Figure 3 also gives pretty much the same information as figure 2.

Author Response

Reviewer #1

Second thing that generally came to my mind – whe did the Authors use only two dosages of AsA? In most cases the higher dose gave better results, so it just asks for checking how far would that go. It should have a maximum level at which it helps, and then too concentrated AsA spray should rather harm the plants, at least I imagine it like that. Why was it not checked? Why wasn’t the optimal dosages established? I think that this study is incomplete without this.

Authors’ Response:

We checked the available literature first before planning for this experiment. Besides, we carried out the preliminary experiment to verify the level of AsA we can apply as foliar spraying. From the preliminary experiment, we concluded that the doses up to 400 mg L-1 are not harmful and can enhance plant growth. Therefore, we did not need to use doses higher than 400 mg L-1. Also, the main purpose of this work was to examine the efficiency of foliar-applied AsA for counteracting the drought stress in common bean plants. Hence, pointing the safe level of AsA which brings the significant improvement of plant stress tolerance as compared to the control let to achieve the required target of this study.

Re: Then the information about it should be included in the introduction, or in the Materials and metods section, not only in this response.

Authors’ response:

We added a sentence confirming that in the Materials and Methods section.

Reviewer #1

Second thing about the Tables: you write in the Materials and Methods that there were six treatments, namely T1-T6 and that they are also referred to as WWAsA or WSAsA (1-3). And then we can see in the Tables results for WW, WS, AsA1-3 separately, and T1-T6 separately. What is WW then if not T1? What is AsA1, 2 and 3 if the foliar spray was always used on WW or WS plants? How can it be singled out? Where those results, that are different from T1-T6, come from?

Authors’ Response:

We have two factors the first water management (WM), which includes two levels i.e. well-watered treatment (WW), and water-stressed treatment (WS). The second factor is the application of ascorbic acid with three ascorbic acid (AsA) treatments, i.e., 0 mg L–1 (AsA1), 200 mg L–1 (AsA2), and 400 mg L–1 (AsA3). From these two factors (2 x 3) we got 6 treatments as a combination of the two factors as mentioned in the text and tables.

Re: Okay, and I understood it that way the forst time I read your MS, but this is not what I was asking about. In Materials and Methods you wrote: „the experiment consisted of six treatments as combinations of two irrigation regimes and three ascorbic acid doses, namely, T1 – 100% ETc 515 with 0 mg L–1 AsA (WWAsA1); T2 – 100% ETc with 200 mg L–1 AsA (WWAsA2); T3 – 100% ETc with 516 400 mg L–1 AsA (WWAsA3); T4 – 50% ETc with 0 mg L–1 AsA (WSAsA1); T5 – 50% ETc with 200 mg 517 L–1 AsA (WSAsA2); and T6 – 50% ETc with 400 mg L–1 AsA (WSAsA3).”

I asked how did you single out only WW, only WS, only AsA1-3? It is inconsistent with Materials and Methods and I don’t know still where it came from. You write that you had 6 treatment combinations and I asked what is AsA1 (alone) if it is now WWAsA1 (T1)? What is WS (alone) if it is not WSAsA1 (T4)? I don’t understand how you got the results for AsA1 that are different than the results for T1. Is AsA1 a mean for T1 and T4 or something? If it is, and you got the results for single factors by any kind of calculations, than it should be stated clearly in the text, also in the tables’ descriptions.

Authors’ response:

Please, note that WW and WS are level 1 and level 2, respectively of factor water management (WM). AsA1 is level 1 of ascorbic acid namely 0 mg L–1 AsA. Yes, WWAsA1 is T1. Yes, WS alone is level 2 from WM but WSAsA1 is T4. AsA1 is level 1 of ascorbic acid and not a mean of T1 and T4.

There are differences between factors and levels of each factor and treatments which came from a combination of both 2 factors (WM & AsA).

Reviewer #1

Line 180-181: Where are those results shown? Table 4 only gives us the level of those compounds 46 days after sowing. Was it WW or WS? Was it treated with AsA? We do not know.

Authors’ Response:

Due to financial constraints to analyse several samples by HLPC, we analysed only the treatment which gave the most evident results on phenolic profile for common bean, i.e., T3, 100% ETc with 400 mg L–1 AsA.

Re: it should be indicated in the Materials and Methods, that it was only measured for this time point then.

Authors’ response:

We indicated that in the Materials and Methods.

Reviewer #1

Table 8: You write in the Table’s title that it refers to two WMs and three AsA treatments – but it is not shown in that Table. We can see various parameters but it is unclear when they were measured, in what conditions. The results certainly show one conditio though, not six, as described.

Authors’ Response:

The correlation matrix is based on means. The number of means in the present experiment is 6: 2 factors with 3 levels = 6 treatments.

Re: So you show a mean for all 6 treatments for one parameter? For example chlb in this table is a mean value received for chlb content in all T1-T6 treatments? If so, then it should be clearly stated in the description of the table, and in Materials and Methods, so that you will not be misunderstood (like you were now by me). Secondly I do not see why the authors put those results in this study. They are interesting but the correlations between various parameters without context (the context was lost by calculating the mean for all of the treatments, now this is only a correlation between those parameters in bean) should be put in a different study, with different research hypothesis.

Authors’ response:

Thank you for your note. We indicated that in the Results section at the footnote of Table 8 and we used the correlation matrix to summarize our data. The correlation matrix table showed the correlation coefficients between 17 traits. Each random trait (Xi) in the table is correlated with each of the other values in the table (Xj). This allows us to see which pairs have the highest correlation.

Reviewer #1

Paragraph 2.11 (Lines 273-286): I cannot see why it was done. These results have almost no relations to what was in the rest of the study and brings nothing to the conclusions. It looks like it was done just because it could be done and to increase the volume of the MS.

Authors’ Response:

The cluster analysis is important to explain or results. It is a technique to group similar observations into some clusters based on the observed values of several variables for each individual.

Re: I know what cluster analysis is, and what it is used for. I just don’t see the connection between the aim of this study, your research hypothesis, and showing cluster analysis for the parameters if it is done on means from all of the treatments. You lose the connection to your study in that way and get the results which are interesting, but I do not see how they are important to explain your results. They would be if they were shown separately for each treatment x treamtment correlation

Authors’ response:

Thank you for your note. We already removed the cluster analysis from the last revision and we left the correlation matrix table.

Reviewer #1

Paragraph 2.13 (Lines 320-338): First of all, this is a presentation of results that were alreade presented in a Table, which even the Authors admit. It is not in accordance with the rules of writing a scientific manuscript!

Authors’ Response:

We do not agree with the Reviewer’s opinion. Paragraph 2.13 (Lines 320-338), presents a vector view of the interrelationship among measured traits for six treatments resulting from the combination of water management and ascorbic acid, while Tables present the mean of each trait.

Re: Yes, the Table presents the correlations (interrelationship) between means of the measured traits, and the Figure presents a vector view of the interrelationship (correlation) among means of the measured traits – I still think you should only pick one. You can observe that the parameters with a correlation close to 1 are very close to each other in this figure etc. Figure 3 also gives pretty much the same information as figure 2.

Authors’ response:

Based on your opinion we removed Fig. 3 and 2.13 section from Results and left Fig. 2.

Reviewer 3 Report

Dear Authors,

As I stated in my previous review, the experimental layout contains serious mistakes which have not been excluded in the revised version of the manuscript.

I cannot accept this manuscript under any conditions.

Author Response

Reviewer #3_R2

Dear Authors,

As I stated in my previous review, the experimental layout contains serious mistakes which have not been excluded in the revised version of the manuscript. I cannot accept this manuscript under any conditions.

Authors’ response:

Please, note that the experimental layout is clear. Please, refer to our response to R1 review, we addressed all your comments.

The experiment was conducted with a split-plot design in a randomized complete block design in three replicates. The main plots included two water managements, namely, 1) 100% of crop evapotranspiration (ETc) throughout the season as a well-watered treatment (WW), and 2) 50% of ETc throughout the season as a water-stressed treatment (WS). The sub-plots were assigned to three ascorbic acid (AsA) treatments, i.e., 1) 0 mg L–1 AsA, 2) 200 mg L–1 AsA, and 3) 400 mg L–1 AsA.

In this experiment we had two factors water management (WM) and Ascorbic acid (AsA). From combination of these factors we got 6 treatments as 2 WM x 3 AsA = 6 treatments. The six treatments namely, T1, 100% ETc with 0 mg L–1 AsA (WWAsA1); T2, 100% ETc with 200 mg L–1 AsA (WWAsA2); T3, 100% ETc with 400 mg L–1 AsA (WWAsA3); T4, 50% ETc with 0 mg L–1 AsA (WSAsA1); T5, 50% ETc with 200 mg L–1 AsA (WSAsA2); and T6, 50% ETc with 400 mg L–1 AsA (WSAsA3).